# *Drosophila* mushroom bodies integrate hunger and satiety signals to control innate food-seeking behavior

Chang-Hui Tsao[1], Chien-Chun Chen[1], Chen-Han Lin[1,2], Hao-Yu Yang[1], Suewei Lin[1,2]*

[1]Institute of Molecular Biology, Academia Sinica, Taipei, Taiwan; [2]Department of Life Sciences and the Institute of Genome Sciences, National Yang-Ming University, Taipei, Taiwan

**Abstract** The fruit fly can evaluate its energy state and decide whether to pursue food-related cues. Here, we reveal that the mushroom body (MB) integrates hunger and satiety signals to control food-seeking behavior. We have discovered five pathways in the MB essential for hungry flies to locate and approach food. Blocking the MB-intrinsic Kenyon cells (KCs) and the MB output neurons (MBONs) in these pathways impairs food-seeking behavior. Starvation bi-directionally modulates MBON responses to a food odor, suggesting that hunger and satiety controls occur at the KC-to-MBON synapses. These controls are mediated by six types of dopaminergic neurons (DANs). By manipulating these DANs, we could inhibit food-seeking behavior in hungry flies or promote food seeking in fed flies. Finally, we show that the DANs potentially receive multiple inputs of hunger and satiety signals. This work demonstrates an information-rich central circuit in the fly brain that controls hunger-driven food-seeking behavior.

DOI: https://doi.org/10.7554/eLife.35264.001

*For correspondence:
sueweilin@gate.sinica.edu.tw

Competing interests: The authors declare that no competing interests exist.

## Introduction

Searching for food is costly with respect to energy and physical risk. Most animals are equipped with an ability to evaluate their internal energy state and use it to decide whether to respond to food-related cues such as taste and smell. To achieve this, the nervous system of an animal must sense its energy state, produce signals that reflect the energy status, and integrate the signals with external sensory inputs. Understanding the computational and operational principles that underpin these neural processes will offer insights into the neural basis of motivated behaviors. The main nutrient- and energy-sensing organs in vertebrates are stomach, gut, and white adipose tissue (*Kairupan et al., 2016*; *Porte et al., 2002*; *Small and Bloom, 2004*). Hormones such as leptin, ghrelin, neuropeptide Y (NPY), and cholecystokinin that are secreted by these organs serve as hunger and satiety signals in the nervous system and control how an animal responds to food cues (*Kairupan et al., 2016*; *Porte et al., 2002*; *Sternson et al., 2013*; *Sternson and Eiselt, 2017*). The detailed molecular and cellular mechanisms involved in this process remains to be elucidated.

Hormonal signals also play important roles in mediating hunger control in the fruit fly *Drosophila*; these include insulin-like peptides, the two homologs of mammalian NPY (Neuropeptide F, NPF, and short Neuropeptide F, sNPF), the homolog of mammalian leptin (Unpaired 2, Upd2), the insect analog of glucagon (adipokinetic hormone, AKH), and a handful of other neuropeptides and metabolites (*Dus et al., 2015*; *Inagaki et al., 2014*; *Jourjine et al., 2016*; *Kim et al., 2017*; *Lee et al., 2004*; *Pool and Scott, 2014*; *Rajan and Perrimon, 2012*; *Root et al., 2011*; *Sun et al., 2017*; *Wu et al., 2005*; *Yu et al., 2016*). These hormonal signals and neuromodulators are regulated by starvation and have been shown to modulate neural circuit functions in both the periphery

(*Farhan et al., 2013*; *Inagaki et al., 2014*; *2012*; *Ko et al., 2015*; *LeDue et al., 2016*; *Root et al., 2011*) and the brain (*Beshel et al., 2017*; *Beshel and Zhong, 2013*; *Schlegel et al., 2016*; *Wang et al., 2013*; *Yu et al., 2016*). Flies primarily rely on olfactory cues to locate food. Starvation sensitizes the olfactory receptor neurons (ORNs) for attractive odors via the cooperation of insulin and sNPF signaling pathways (*Root et al., 2011*) and, in parallel, dampens the activity of ORNs for aversive odors through the neuropeptide tachykinin released from interneurons in the antennal lobe (*Ko et al., 2015*). Another neuropeptide, CCHamide, is also involved in hunger-induced modulation in ORNs (*Farhan et al., 2013*). Whether a similar system exists in the brain to regulate the perception of odor valence in accordance with the hunger state is less clear. A pair of NPF-expressing neurons in the adult brain have been shown to encode odor attractiveness (*Beshel and Zhong, 2013*). More attractive odors evoke stronger activity in the NPF neurons, and silencing of these neurons abolishes the fly's behavioral response to attractive odors. Importantly, starvation heightens the activity of the NPF neurons, but how the graded NPF signal is translated into approach behavior is not known.

Food odors detected by specific ORNs are relayed by antennal lobe projection neurons to the mushroom body (MB) and lateral horn (LH) (*Jefferis et al., 2007*; *Lin et al., 2007*; *Stocker et al., 1997*). It has been suggested that the LH plays a major role in innate olfactory behavior (*de Belle and Heisenberg, 1994*; *Heimbeck et al., 2001*; *Jefferis et al., 2007*; *Parnas et al., 2013*; *Strutz et al., 2014*). Neural modulation in the LH has been shown to regulate feeding behavior in *Drosophila* larvae. Brief presentation of appetizing odors causes voracious feeding on sugar-rich food even in fed larvae; a process involving NPF neurons and dopaminergic neurons projecting to the LH (*Wang et al., 2013*). Whether the same neural pathways mediate starvation-induced feeding awaits to be tested. In contrast to the LH, the MB is conventionally considered to be an olfactory learning and memory center (*de Belle and Heisenberg, 1994*; *Heisenberg, 2003*; *Keene and Waddell, 2007*; *McGuire et al., 2005*). The MB is composed of around 2200 intrinsic neurons called Kenyon Cells (KC), which extend parallel axonal fibers to form the γ, α'β', and αβ lobes of the MB (*Aso et al., 2014a*; *Crittenden et al., 1998*; *Lin et al., 2007*; *Strausfeld et al., 2003*; *Tanaka et al., 2008*). Each KC receives inputs from a random combination of antennal lobe projection neurons (*Caron et al., 2013*). The KC outputs converge onto 21 types of 34 MB output neurons (MBONs). The dendrites of each MBON type arborize in specific compartments of the MB lobes, and all MBON types together innervate 15 distinct compartments that tile the MB lobes (*Aso et al., 2014a*). The MB lobes are also extensively innervated by dopaminergic neurons (DANs). About 130 MB-innervating DANs of 20 cell types have been identified (*Aso et al., 2014a*). Like the MBONs, each DAN type projects its axons to distinct MB lobe zones. Distinct types of DANs react selectively to punishment or reward stimuli to potentiate or depress KC-to-MBON synapses in specific compartments; a process believed to be the neural basis of olfactory associative learning (*Aso et al., 2012*; *Aso and Rubin, 2016*; *Burke et al., 2012*; *Das et al., 2014*; *Galili et al., 2014*; *Hige et al., 2015a*; *Huetteroth et al., 2015*; *Lin et al., 2014b*; *Liu et al., 2012*; *Owald et al., 2015*; *Yamagata et al., 2015*). The role of the MB in innate olfactory behavior has been less studied. It was initially reported that the MB is required for innate odor attraction but not repulsion (*Wang et al., 2003*). However, recent studies suggest that the MB circuit also regulates innate odor repulsion (*Bräcker et al., 2013*; *Lewis et al., 2015*; *Owald et al., 2015*; *Perisse et al., 2016*; *Siju et al., 2014*). Blocking the MBONs innervating the tips of the horizontal MB lobes impairs a fly's response to aversive odors (*Lewis et al., 2015*; *Owald et al., 2015*). It has been proposed that starvation controls sugar memory expression by modulating odor-evoked responses in these MBONs via other neurons in the MB circuit (*Perisse et al., 2016*). The same MBONs also respond to $CO_2$ and mediate $CO_2$-induced avoidance behavior in both fed and starved flies (*Lewis et al., 2015*). However, blocking neurotransmission of the KCs impairs $CO_2$ avoidance only in hungry flies, but not in fed flies (*Bräcker et al., 2013*). Apparently, hunger and satiety states can influence the information processing of the MB, but to what extent and whether the MB circuit is utilized to regulate hunger-evoked food-seeking behavior in naive flies remains to be addressed.

In this study, we show that the MB plays an essential role in controlling food-seeking behavior. Flies approach yeast food only when they are hungry. Blocking the KCs strongly impairs yeast food-seeking behavior in hungry flies. We have identified five MBONs required for hungry flies to seek not only yeast odor, but also apple cider vinegar and banana odors. In vivo functional imaging showed that these MBONs respond to yeast odor and that the responses were modulated by

starvation. There are six types of DANs innervating the same MB lobe compartments occupied by the dendritic arbors of the five MBONs. Blockage and activation of these DANs inhibited and promoted yeast food-seeking behavior, respectively. The function of the DANs is mediated by the dopamine receptor DAMB in both the KCs and the MBONs. Finally, we demonstrate that the six DANs potentially receive rich and diverse inputs of hunger and satiety signals. Our data establish the MB as an integration center for hunger-control of innate food-seeking behavior. The DANs constantly monitor the metabolic state of the fly and, when there is a need, reconfigure the KC-to-MBON circuits in specific MB lobe compartments to bias the fly's response to food odors.

## Results

### Flies approach yeast food when they are hungry

We used a single-fly assay to quantify each fly's food-seeking behavior. For each assay, a male fly was allowed to move freely in a petri dish for 10 min to locate a drop of yeast. Yeast is an ethologically relevant food source for flies and yeast odor has been shown to be a strong attractant for them (*Scheidler et al., 2015*; *Stökl et al., 2010*). The assay was performed under red light (630 nm) to minimize visual inputs to the fly. To avoid scoring flies that accidentally passed by the yeast drop, we considered that the fly had found the target when it located and remained on the yeast drop for 3 s or more. We calculated each fly's food-seeking performance based on how quickly it found the yeast drop (see Materials and methods for details). At the population level, average yeast food-seeking performance increased linearly with the duration of food deprivation, suggesting that hunger regulates food-seeking behavior in a graded manner within a high dynamic range (*Figure 1A*). Flies starved for 24 hr did not seek a water-only drop, confirming that they were attracted to yeast in this assay (*Figure 1A*). To determine whether flies used olfactory cues to locate food in our assays, we tested mutant flies lacking co-receptors for odorant receptors (ORs) and ionotropic receptors (IRs). We found that *orco* and *Ir8a* mutant flies could not locate yeast drops (*Figure 1B*), indicating that flies relied on their sense of smell to find the target. This result is consistent with previous findings that flies use both OR and IR systems to smell yeast odor (*Gorter et al., 2016*; *Libert et al., 2007*).

### Kenyon cells are crucial for yeast food-seeking behavior in hungry flies

Flies detect odors by means of the sensory neurons in their antennae. The olfactory information is then relayed to the MB and LH by antennal lobe projection neurons. We next examined whether the MB is required for hungry flies to follow yeast odor. We blocked neurotransmission with *UAS-shi^{ts1}*—a temperature-sensitive dominant-negative dynamin transgene (*Kitamoto, 2001*)—in different types of KCs and tested the flies' yeast food-seeking performance at the restrictive temperature of 32°C. All three major types of KCs—γ, α′β′, and αβ neurons labeled by *MB131B-splitGAL4*, *MB005B-splitGAL4*, and *MB008B-splitGAL4*, respectively—were found to be important for yeast food-seeking behavior (*Figure 2*). When these neurons were blocked, the fly took longer to locate the food, and many flies walked randomly as if they were insensitive to yeast odor. Flies whose KCs were blocked did not show any sign of locomotion defects and they performed normally when assayed at the permissive temperature of 23°C (*Figure 2—figure supplement 1*).

### Five MBON pathways are required for food-seeking behavior in hungry flies

To identify the output channels in the MB circuits that promote yeast food-seeking behavior, we screened 36 GAL4 and split-GAL4 lines that cover all of the 22 MBON types (*Figure 3—figure supplement 1*) (*Aso et al., 2014b*). We uncovered six MBON types whose blockage by *UAS-shi^{ts1}* caused yeast food-seeking defects in hungry flies (*Figure 3* and *Figure 3—figure supplement 1*). It is noteworthy that blocking neurotransmission of the MBONs has been shown not to cause measurable locomotion defects (*Aso et al., 2014b*). We further confirmed this by measuring the moving speed of hungry flies whose MBONs for yeast food-seeking were blocked (*Figure 3—figure supplement 2*).

The six MBON types we identified in our screen were MBON-γ1pedc>αβ, MBON-β2β′2a, MBON-γ2α′1, MBON-α′2, MBON-α3, and MBON-β1>α. Several split-GAL4 lines specifically labeled MBON-γ1pedc>αβ, MBON-β2β′2a, MBON-γ2α′1, and MBON-α′2 (*Figure 3A–D*). Using these drivers to

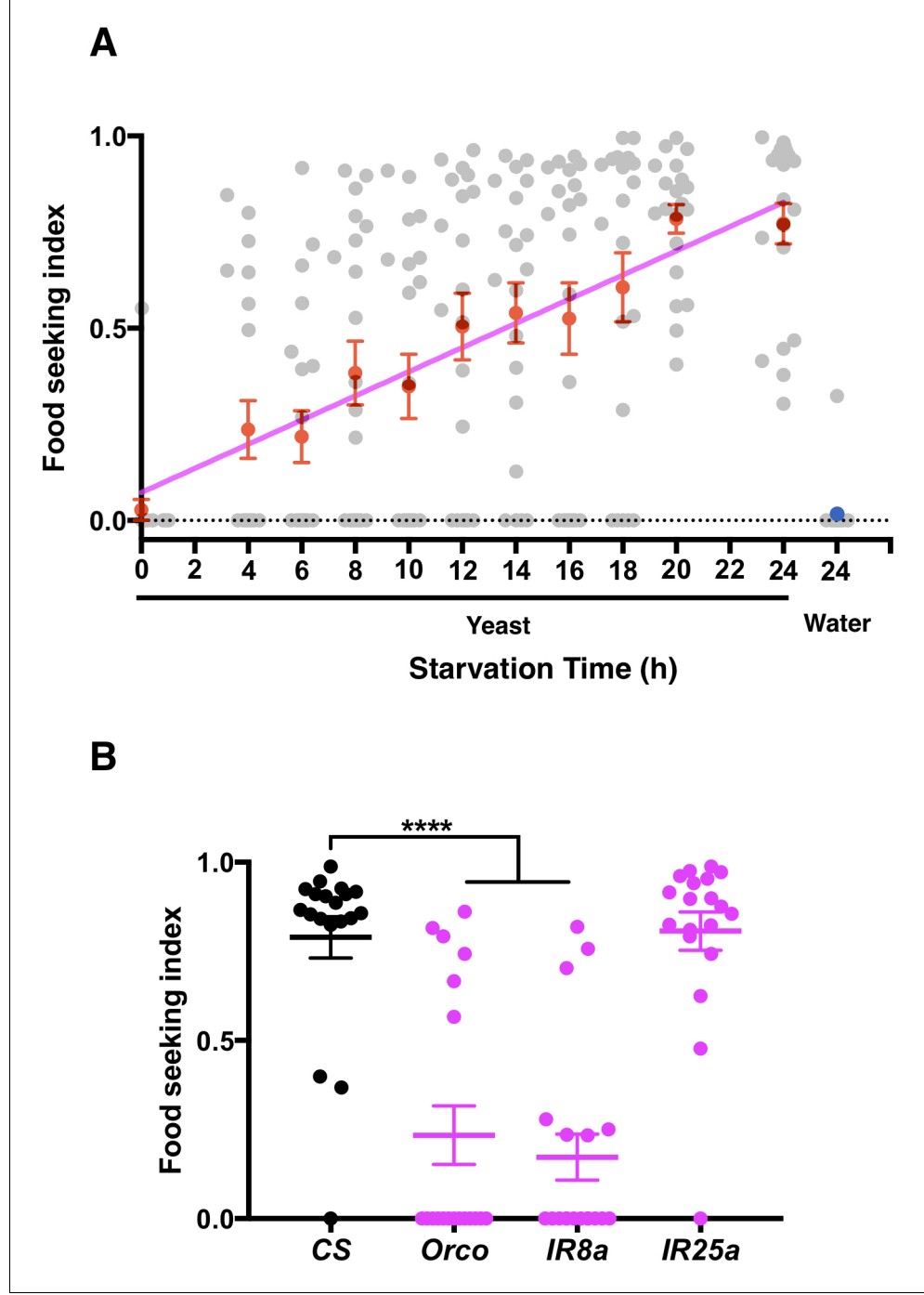

**Figure 1.** Starvation promotes yeast food-seeking behavior. (**A**) The average yeast food-seeking performance (y-axis; see Materials and methods for mode of calculation) increases linearly with the duration of starvation (x-axis). A water-only control for flies starved for 24 hr is also shown. Individual data points and mean ± SEM (n = 20 for each point) are shown. (**B**) The yeast food-seeking performance of wild-type flies (CS) and flies homozygous for *orco²*, *IR8a¹*, and *IR25a²*. The performances of the *orco²* and *IR8a¹* flies were significantly lower than that of the control flies (Kruskal-Wallis, n = 19, p<0.0001). Individual data points and mean ± SEM are shown.
DOI: https://doi.org/10.7554/eLife.35264.002

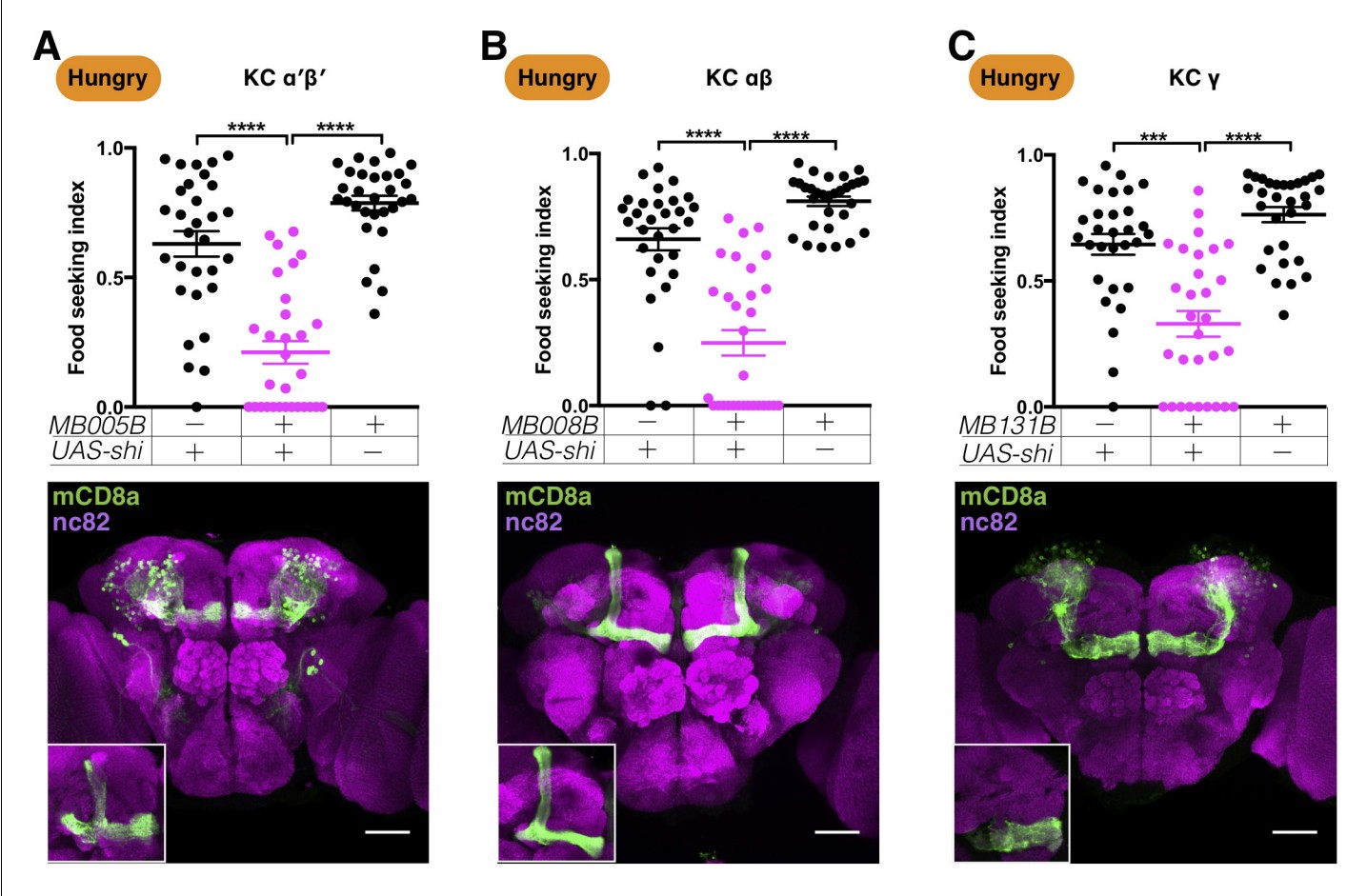

**Figure 2.** Kenyon cells are required for yeast food-seeking behavior in hungry flies. (A–C) Male flies starved for 24 hr were assessed for their yeast food-seeking performance. The performance of *GAL4;UAS-shi^ts1^* flies was statistically different from the controls for (A) *MB005B split-GAL4* (α′β′ KCs; Kruskal-Wallis, n = 30, p<0.0001), (B) *MB008B split-GAL4* (αβ KCs; Kruskal-Wallis, n = 30, p<0.0001), and (C) *MB131B split-GAL4* (γ KCs; Kruskal-Wallis, n = 30, p=0.0003). Individual data points and mean ± SEM are shown. The brain images are full z-projections of confocal stacks showing the expression patterns of the GAL4 lines (green) counter-stained with nc82 antibody (magenta). Insets are z-projections of the MB lobes. Scale bars are 100 μm.
DOI: https://doi.org/10.7554/eLife.35264.003

The following figure supplement is available for figure 2:

**Figure supplement 1.** Expression of *UAS-shi^ts1^* in the KCs does not affect yeast food-seeking behavior in hungry flies at the permissive temperature.
DOI: https://doi.org/10.7554/eLife.35264.004

express *UAS-shi^ts1^* significantly impaired hungry flies' yeast-seeking performance at the restrictive temperature (*Figure 3A–D*), but not at the permissive temperature (*Figure 3—figure supplement 3A–D*). Two split-GAL4 lines, *MB082C* and *MB093C*, labeled both MBON-α′2 and MBON-α3 (*Aso et al., 2014b*), and blocking neurotransmission with these two lines and *UAS-shi^ts1^* resulted in yeast-seeking defects (*Figure 3—figure supplement 1*). To examine whether MBON-α3 plays a role in controlling yeast food-seeking behavior, we used two additional GAL4 lines, *G0239* and *E0067*, that specifically label MBON-α3 (*Pai et al., 2013*; *Plaçais et al., 2013*). Blocking the neurotransmission of MBON-α3 alone was sufficient to impair yeast food-seeking behavior (*Figure 3—figure supplement 1*; *Figure 3E* and *Figure 3—figure supplement 1E*). Thus, both MBON-α′2 and MBON-α3 are required for hungry flies to seek yeast food. MBON-β1>α was labeled by the split-GAL4 lines *MB433B* and *MB434B*. Using these two lines to drive *UAS-shi^ts1^* impaired yeast food-seeking behavior at the restrictive temperature (*Figure 3—figure supplement 1*). *MB433B* and *MB434B* also label MBON-γ4>γ1γ2. Blocking MBON-γ4>γ1γ2 alone with *MB298B-splitGAL4* and *UAS-shi^ts1^* had no effect on yeast food-seeking behavior in hungry flies (*Figure 3—figure supplement 1*). We could not identify a GAL4 line that would allow us to specifically manipulate MBON-β1>α so we did not

investigate this neuron type further. We got mixed results with the split-GAL4 lines that label MBONs innervating the β′2 regions. *MB011B-splitGAL4* and *MB210B-splitGAL4* essentially label the same β′2-innervating MBON types (*Aso et al., 2014a*; *Aso et al., 2014b*), but *MB011B* with *UAS-shi^{ts1}* resulted in a yeast-seeking defect and flies with *MB210B* and *UAS-shi^{ts1}* performed normally in yeast-seeking behavior at the restrictive temperature (*Figure 3—figure supplement 1*). Similarly, inconsistent results were observed between *MB002B-GAL4* and *VT1211-GAL4*, which label the same MBON types (*Figure 3—figure supplement 1*). Therefore, there is no strong evidence to suggest that these β′2-innervating MBONs are required for hungry flies to seek yeast food. In summary, we have identified five MBON types—MBON-γ1pedc>αβ, MBON-β2β′2a, MBON-γ2α′1, MBON-α′2, and MBON-α3—that are critical for yeast food-seeking behavior in hungry flies. To further ensure that the yeast food-seeking phenotype caused by the blockage of the five MBONs was mainly due to the flies' inability to locate and approach yeast food, we altered how we scored the flies and considered flies as finding the target whenever they touched the yeast drop. Nevertheless, we still found equally strong yeast food-seeking defects using this approach (*Figure 3—figure supplement 4A–E*), which was also the case when we used the same scoring method to test flies with blockage of neurotransmission in three major types of KCs (*Figure 3—figure supplement 4F–H*).

Some MBONs have been shown to encode valence (*Aso et al., 2014b*). Interestingly, not all positive-valence MBONs are required for yeast food-seeking behavior in hungry flies (*Figure 3—figure supplement 1*). Of the five MBONs we identified, only MBON-γ1pedc>αβ and MBON-γ2α′1 encode positive valence and drive approach behavior; the other three MBONs elicit neither approach nor avoidance behavior when optogenetically activated (*Aso et al., 2014b*).

Are the five MBONs specifically required for yeast-odor seeking or are they also required for hungry flies to seek other food odors? To test these possibilities, we blocked the five MBONs and examined hungry flies' performance in seeking the source of apple cider vinegar (ACV) and banana odors. We found that all five MBONs required for yeast food-seeking are also important for hungry flies to seek and approach sources of ACV and banana odor (*Figure 4* and *Figure 4—figure supplement 1*). We also tested the three major KC populations and found that they were also required for ACV- and banana odor-seeking behavior (*Figure 4—figure supplement 2*). These results indicate that the KCs and the five MBONs pathways we identified using the yeast food-seeking assay are generally involved in regulating food odor-seeking behavior in hungry flies.

## The five identified MBONs show modified responses to yeast odor in hungry flies

Since only hungry flies approach yeast food, we expected that starvation would modulate the responses of the five identified MBONs to yeast odor. We mounted individual flies under a two-photon microscope and presented them with a yeast odor stimulus for 10 s. Odor-evoked calcium transients in the MBONs were imaged using the genetically-encoded calcium indicator GCaMP6m (*Chen et al., 2013*). All five MBONs showed hunger-dependent changes in their responses to yeast odor (*Figure 5*). Interestingly, starvation modulated the two positive-valence MBONs contrastingly: it potentiated MBON-γ1pedc>αβ and depressed MBON-γ2α′1 (*Figure 5A and B*). Since blocking MBON-γ2α′1 impairs yeast food-seeking behavior (*Figure 3B*), the diminished response to yeast odor in hungry flies suggests that the level of response might be critical (see Discussion for details). Hunger-dependent potentiation was also observed in MBON-α3 (*Figure 5C*). In MBON-β2β′2a and MBON-α′2, starvation depressed the neurons' response to yeast odor (*Figure 5D and E*). We note that the reduced hunger-dependent odor response for MBON-γ2α′1 and MBON-α′2 was more pronounced for diluted odors (*Figure 5B and E*). Therefore, these neurons are more sensitive to changes in odor concentration (at least for the concentration range that we tested) when the fly is starved. This modulation in odor response may be important for hungry flies to navigate along an odor concentration gradient to locate food. Together, our findings show that starvation changes how yeast odor information is processed in the MB circuit by fine-tuning the activities of the five MBONs required for yeast food-seeking behavior. Since an overall increase or decrease in the responses of KCs to yeast odor should result in a general increase or decrease in odor-evoked responses in the MBONs, the bi-directional tuning suggests that the starvation-induced modification likely happens at the KC-to-MBON synapses. It has been shown that when food odor-evoked calcium transients are measured from the cell bodies of the entire KC population, no difference was observed between fed and starved flies (*Beshel and Zhong, 2013*). To check whether this is the

case when odor-evoked calcium transients are recorded from individual compartments in the MB lobes, we expressed GCaMP6m in all KCs using *MB010B-splitGAL4* or in γ KCs-only using *MB131B-splitGAL4*, and measured Ca²⁺ signals from the compartments innervated by the five food-seeking MBONs. Consistent with previous findings, the yeast odor-evoked responses in these compartments did not differ between fed and hungry flies (*Figure 5—figure supplement 1*). However, unexpectedly, we found that Ca²⁺ signal in the α′2 compartment decreased in response to yeast odor in both fed and hungry flies (*Figure 5—figure supplement 1E*). The nature of this decreased Ca²⁺ signal and how it is connected to the activation of MBON-α′2 (*Figure 5E*) are unclear. We did not investigate these issues further, but it would be interesting to find out if this decrease in Ca²⁺ signal is specific to yeast odor and if MBON-α′2 also receives inputs from other MB lobe compartments. Overall, our findings and those of others (*Beshel and Zhong, 2013*) suggest that starvation-induced tuning in yeast odor-evoked responses in the five MBONs is not due to changes in the responses of KCs to yeast odor, supporting that the modulations happen at the KC-to-MBON synapses.

## GABAergic inputs in the α1 and β′2 lobe compartments promote yeast food-seeking behavior in hungry flies

MBON-γ1pedc>αβ is GABAergic and has been shown to promote appetitive memory expression by inhibiting the activity of MBONs innervating the β′2 zones (*Aso et al., 2014a*; *Perisse et al., 2016*). We probed this pathway further to assess its role in innate food-seeking behavior. RNAi knockdown of GABA biosynthesis in MBON-γ1pedc>αβ using *MB112C-splitGAL4* and two independent *UAS-GAD-RNAi* lines (*Koganezawa et al., 2016*; *Lin et al., 2014a*) resulted in the same yeast food-seeking defect caused by blocking the neurons' neurotransmission (*Figures 6A* and *3A* and *Figure 6—figure supplement 1A*), suggesting that MBON-γ1pedc>αβ regulates yeast food-seeking behavior through its release of GABA. Knockdown of GABA-A receptors using two independent *UAS-Rdl-RNAi* lines, together with *VT1211-GAL4* labeling of MBON-γ5β′2a and MBON-β′2mp, revealed strong impairment of yeast food-seeking behavior in hungry flies (*Figure 6B* and *Figure 6—figure supplement 1B*). Furthermore, when *UAS-shi^ts1* was expressed with *VT1211-GAL4*, the flies exhibited a significant increase in yeast food-seeking behavior when they were food-satiated at a restrictive 32°C, but not at a permissive 23°C (*Figure 6C* and *Figure 6—figure supplement 2A*). These results support the idea that the GABAergic pathway is used to control innate hunger-driven food-seeking behavior.

The axons of MBON-γ1pedc>αβ mainly innervate α and β lobes (*Aso et al., 2014a*). However, in vivo functional imaging studies suggest that MBON-γ1pedc>αβ inhibits MBONs in the β′2 zones, but not MBONs targeting the α2 zones (*Perisse et al., 2016*). MBONs whose dendrites innervate other compartments along the α and β lobes have not been examined. Since blockage of MBONs innervating the α3 zones (MBON-α3) and β2 zones (MBON-β2β′2a) suppressed rather than promoted yeast food-seeking behavior (*Figure 3D and E* and *Figure 3—figure supplement 1*), these MBONs are unlikely to be the targets of MBON-γ1pedc>αβ, at least in terms of regulating the innate behavioral response of flies to yeast odor. Therefore, we tested whether MBON-α1 is a potential target. Knockdown of the GABA-A receptor with two independent RNAi lines and *MB310C-splitGAL4* that specifically labels MBON-α1 resulted in decreased yeast food-seeking behavior in hungry flies (*Figure 6D* and *Figure 6—figure supplement 1C*). Consistently, blocking the neurotransmission of MBON-α1 with *MB310C-splitGAL4* and *UAS-shi^ts1* promoted yeast food-seeking behavior in fed flies at a restrictive 32°C, but not at a permissive 23°C (*Figure 6E* and *Figure 6—figure supplement 2B*). These results suggest that when flies are starved, increased GABA release from MBON-γ1pedc>αβ inhibits MBON-γ5β′2a/MBON-β′2mp and MBON-α1 via the GABA-A receptor. In turn, suppression of MBON-γ5β′2a/MBON-β′2mp and MBON-α1 positively biases flies' responses toward yeast odor (*Figure 6F*). However, it is important to note that potential involvement of other GABAergic neurons of the MB circuit, such as APL neurons (*Liu and Davis, 2009*), in the regulation of MBON-γ5β′2a/MBON-β′2mp and MBON-α1 cannot be excluded based on our experiments. Furthermore, the direct functional connectivity between MBON-γ1pedc>αβ and MBON-α1 remains to be established.

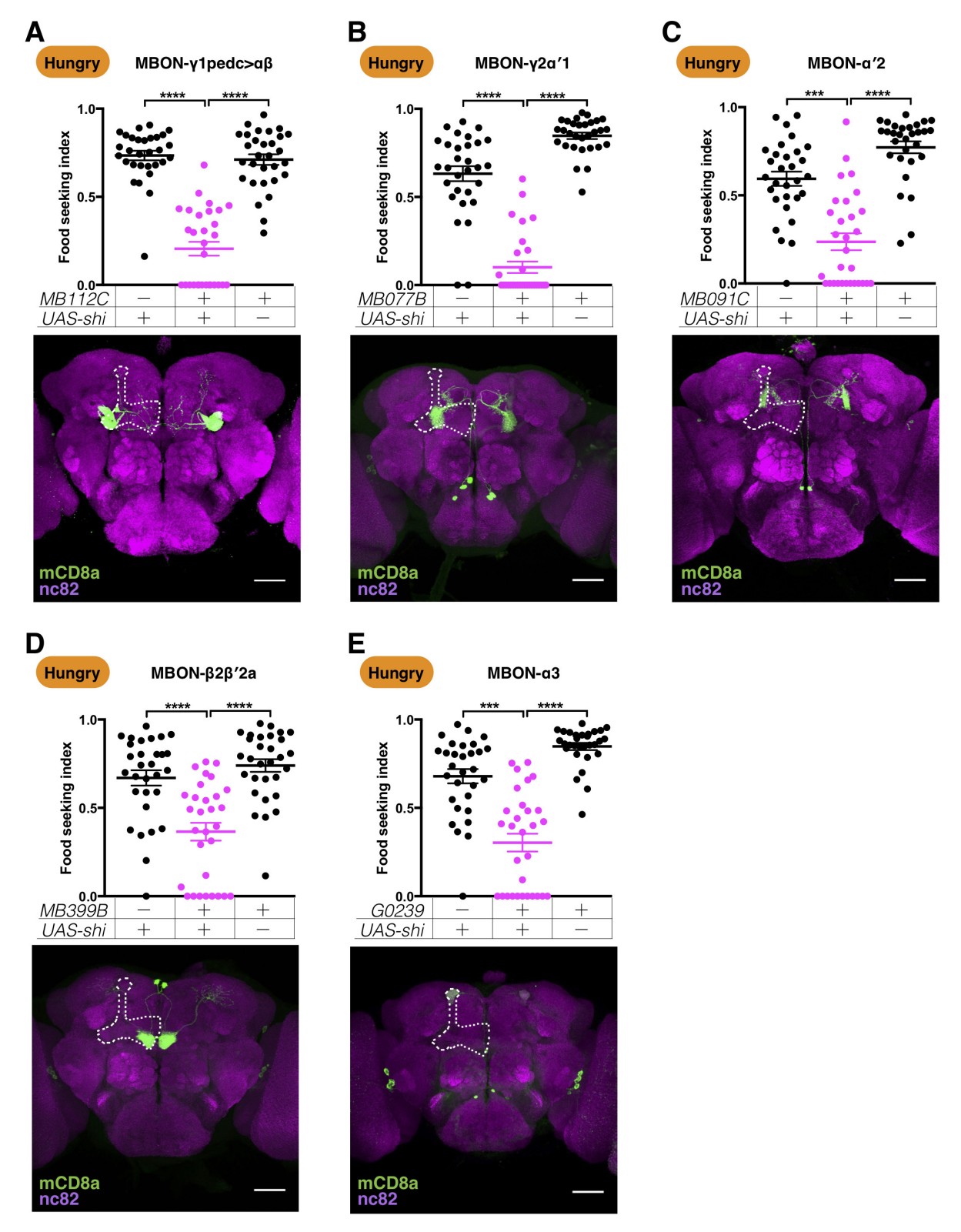

**Figure 3.** Five MBONs are required for yeast food-seeking behavior in hungry flies. Male flies starved for 24 hr were assessed for the yeast food-seeking performance. The performance of *GAL4;UAS-shi^ts1* flies was statistically lower than the controls for (**A**) *MB112C split-GAL4* (MBON-γ1pedc>αβ, Kruskal-Wallis, n = 30, p<0.0001), (**B**) *MB077B split-GAL4* (MBON-γ2α'1, Kruskal-Wallis, n = 30, p<0.0001), (**C**) *MB091C split-GAL4* (MBON-α'2, Kruskal-Wallis, n = 30, p=0.0004), (**D**) *MB399B split-GAL4* (MBON-β2β'2a, Kruskal-Wallis, n = 30, p<0.0001), and (**E**) *G0239-GAL4* (MBON-α3, Kruskal-Wallis, 

*Figure 3 continued on next page*

*Figure 3 continued*

n = 30, p=0.0002). Individual data points and mean ± SEM are shown. The brain images are full z-projections of confocal stacks showing the expression patterns of the GAL4 lines (green) counter-stained with nc82 antibody (magenta). One side of the MB is outlined by a white dashed line. Scale bars are 100 μm.

DOI: https://doi.org/10.7554/eLife.35264.005

The following source data and figure supplements are available for figure 3:

**Figure supplement 1.** Effects on yeast food-seeking behavior in hungry flies when GAL4 lines labeling different MBONs are used to drive *UAS-shi^ts1* expression.

DOI: https://doi.org/10.7554/eLife.35264.006

**Figure supplement 1—source data 1.** Source file for the table in *Figure 3—figure supplement 1*.

DOI: https://doi.org/10.7554/eLife.35264.007

**Figure supplement 2.** Blocking the MBONs required for yeast food-seeking behavior does not affect the locomotion of flies.

DOI: https://doi.org/10.7554/eLife.35264.008

**Figure supplement 3.** Expression of *UAS-shi^ts1* in the five MBONs does not affect yeast food-seeking behavior in hungry flies at the permissive temperature.

DOI: https://doi.org/10.7554/eLife.35264.009

**Figure supplement 4.** The five MBONs and the KCs are required during the seeking phase in our food-seeking assay.

DOI: https://doi.org/10.7554/eLife.35264.010

## Dopaminergic neurons convey hunger to the MB circuit

Our in vivo imaging data suggest that starvation likely modulates the KC-to-MBON synapses. Multiple lines of evidence have suggested that KC-to-MBON connectivity can be shaped by DANs innervating the same MB lobe compartments where the KC and MBON neurons meet (*Aso and Rubin, 2016*; *Cohn et al., 2015*; *Hige et al., 2015a*; *Musso et al., 2015*; *Owald et al., 2015*). Therefore, we examined the corresponding DANs of the yeast-seeking MBONs that we had identified. For MBON-γ1pedc>αβ, MBON-γ2α′1, MBON-α′2, and MBON-α3, there is one DAN type—PPL1-γ1pedc, PPL1-γ2α′1, PPL1-α′2α2, and PPL1-α3, respectively—whose axonal termini overlap with the dendrites of each of the MBON types (*Aso et al., 2014a*). In contrast, MBON-β2β′2a is potentially regulated by two types of DANs: PAM-β2β′2a and PAM-β′2a (*Aso et al., 2014a*). Blockage of the neurotransmission of PPL1-α3, PAM-β2β′2a, PAM-β′2a, PPL1-α′2α2, and PPL1-γ2α′1 DANs with *UAS-shi^ts1* strongly impaired the performances of hungry flies in yeast food-seeking behavior at a restrictive 32°C, but not at a permissive 23°C (*Figure 7A–E* and *Figure 7—figure supplement 1A–E*). These results suggest that the corresponding DANs for MBON-α3, MBON-β2β′2a, MBON-α′2, and MBON-γ2α′1 play important roles in regulating yeast food-seeking behavior and that hunger promotes dopamine release in these DANs. The PPL1-γ1pedc DANs have been shown to repress the activity of MBON-γ1pedc>αβ, thereby mediating the hunger control of sugar memory expression (*Perisse et al., 2016*). Consistently, artificial activation of the PPL1-γ1pedc DANs with *UAS-TrpA1*, a heat-sensitive cation channel transgene (*Hamada et al., 2008*), reduced yeast food-seeking behavior in hungry flies at a restrictive 32°C (*Figure 7F*), but not at a permissive 23°C (*Figure 7—figure supplement 1F*). These results indicate that the fruit fly uses the same PPL1-γ1pedc-to-MBON-γ1pedc>αβ neural pathway to mediate hunger control for both innate and learned food-related olfactory cues and that hunger inhibits rather than promotes the release of dopamine by PPL1-γ1pedc DANs. It remains to be determined whether PPL1-α3, PAM-β2β′2a, PAM-β′2a, PPL1-α′2α2, and PPL1-γ2α′1 DANs play a role in regulating hunger-dependent sugar memory expression.

Blockage of PPL1-γ1pedc DANs is sufficient to promote sugar memory expression (*Krashes et al., 2009*). Indeed, blocking the neurotransmission of PPL1-γ1pedc DANs with *UAS-shi^ts1* also promotes yeast food-seeking behavior in fed flies at a restrictive 32°C (*Figure 7G*), but not at a permissive 23°C (*Figure 7—figure supplement 1G*). We then tested whether artificial activation of PPL1-α3, PAM-β2β′2a, PAM-β′2a, PPL1-α′2α2, and PPL1-γ2α′1 DANs promotes yeast food-seeking behavior. Strikingly, in all cases, driving the expression of *UAS-TrpA1* with split-GAL4 lines that specifically label these DANs made flies approach yeast food even when they were well fed (*Figure 7H–L* and *Figure 7—figure supplement 1H–L*). Pairing odors with artificial activation or silencing of some DANs has been shown to induce positive or negative olfactory memories in the fly (*Aso et al., 2012*; *2010*; *Aso and Rubin, 2016*; *Burke et al., 2012*; *Huetteroth et al., 2015*; *Ichinose et al., 2015*; *Lin et al., 2014b*; *Liu et al., 2012*; *Shyu et al., 2017*; *Yamagata et al., 2015*;

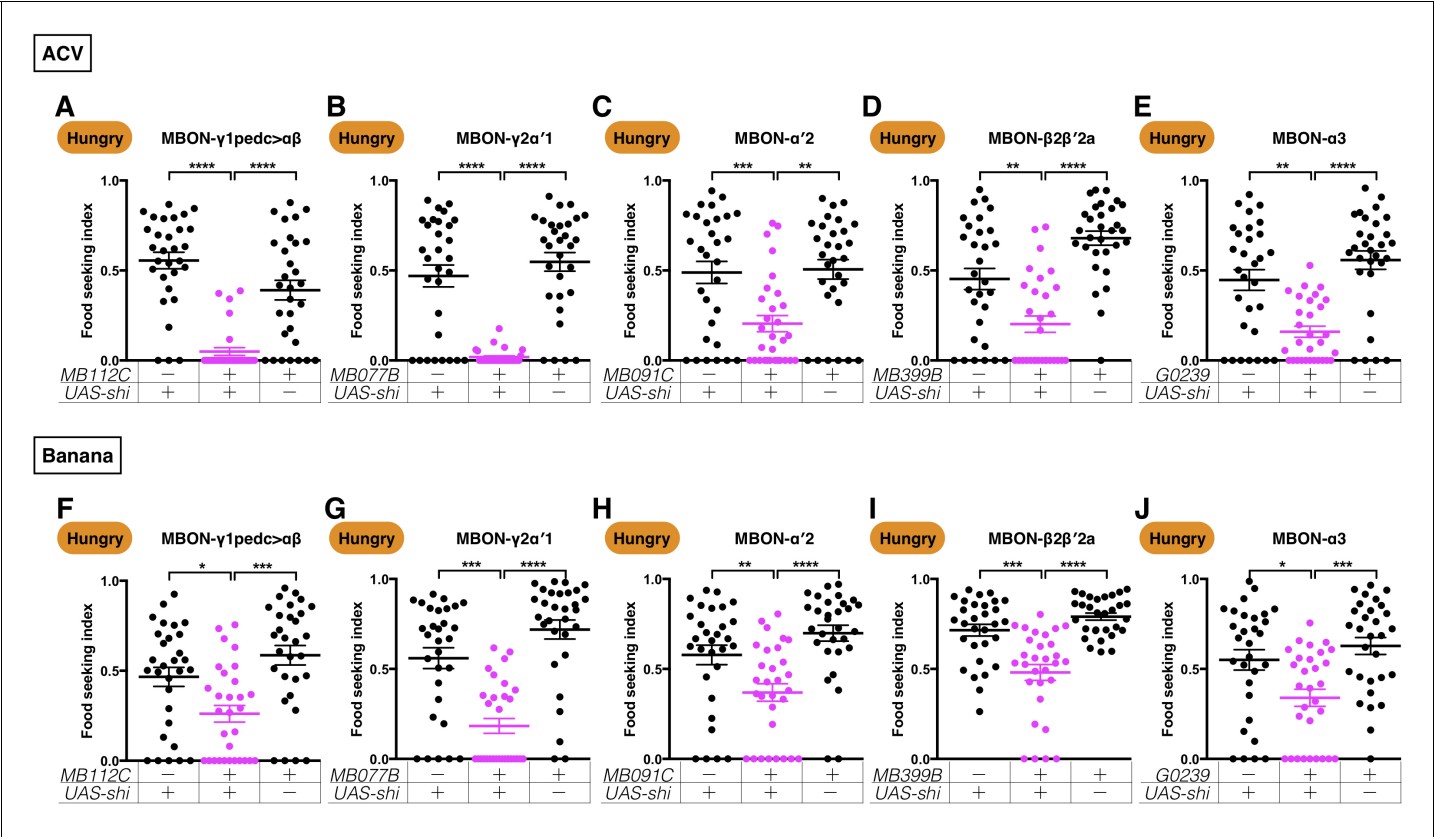

**Figure 4.** Five MBONs are required for hungry flies to seek ACV and banana odors. Male flies starved for 24 hr were assessed for their performance in seeking ACV (**A–E**) or banana odor (**F–J**) at a restrictive 32°C. The performance of *GAL4;UAS-shi*[ts1] flies was significantly lower than the controls for (**A**) *MB112C split-GAL4* (MBON-γ1pedc>αβ, Kruskal-Wallis, n = 30, p<0.0001), (**B**) *MB077B split-GAL4* (MBON-γ2α′1, Kruskal-Wallis, n = 30, p<0.0001), (**C**) *MB091C split-GAL4* (MBON-α′2, Kruskal-Wallis, n = 30, p=0.0025), (**D**) *MB399B split-GAL4* (MBON-β2β′2a, Kruskal-Wallis, n = 30, p=0.009), (**E**) *G0239-GAL4* (MBON-α3, Kruskal-Wallis, n = 30, p=0.0016), (**F**) *MB112C split-GAL4* (MBON-γ1pedc>αβ, Kruskal-Wallis, n = 30, p=0.0246), (**G**) *MB077B split-GAL4* (MBON-γ2α′1, Kruskal-Wallis, n = 30, p=0.0004), (**H**) *MB091C split-GAL4* (MBON-α′2, Kruskal-Wallis, n = 30, p=0.0091), (**I**) *MB399B split-GAL4* (MBON-β2β′2a, Kruskal-Wallis, n = 30, p=0.0002), and (**J**) *G0239-GAL4* (MBON-α3, Kruskal-Wallis, n = 30, p=0.0101). Individual data points and mean ± SEM are shown.

DOI: https://doi.org/10.7554/eLife.35264.011

The following figure supplements are available for figure 4:

**Figure supplement 1.** Expression of *UAS-shi*[ts1] in the five MBONs does not affect ACV and banana odor-seeking behavior in hungry flies at the permissive temperature.

DOI: https://doi.org/10.7554/eLife.35264.012

**Figure supplement 2.** KCs are required for hungry flies to seek ACV and banana odors.

DOI: https://doi.org/10.7554/eLife.35264.013

*Yamagata et al., 2016*). However, the behavioral phenotypes we observed here are unlikely due to olfactory learning. For PPL1-α3, PPL1-α′2α2, and PPL1-γ2α′1 DANs, pairing the activation of these neurons with odors induces aversive olfactory memories (*Aso and Rubin, 2016*) but, in our assay, activation of these three DANs promotes rather than inhibits yeast odor-seeking (*Figure 7H,I and L*). Activation of PPL1-γ1pedc DANs induced aversive memory when paired with an odor and might cause the yeast food-seeking impairment we observed (*Figure 7F*). To investigate this further, we conditioned flies by pairing yeast odor with the activation of PPL1-γ1pedc DANs for 2 min at 32°C and tested their yeast-seeking performance shortly thereafter at 23°C (*Figure 7—figure supplement 2A*). However, these flies performed normally in seeking yeast food, suggesting that olfactory learning contributes minimally in our experiments (*Figure 7—figure supplement 2B*). The minimum olfactory learning effect might be due to yeast odor having a strong innate value to the fly, so it cannot be easily conditioned. Another possible reason is that we activated or silenced DANs for 10 min

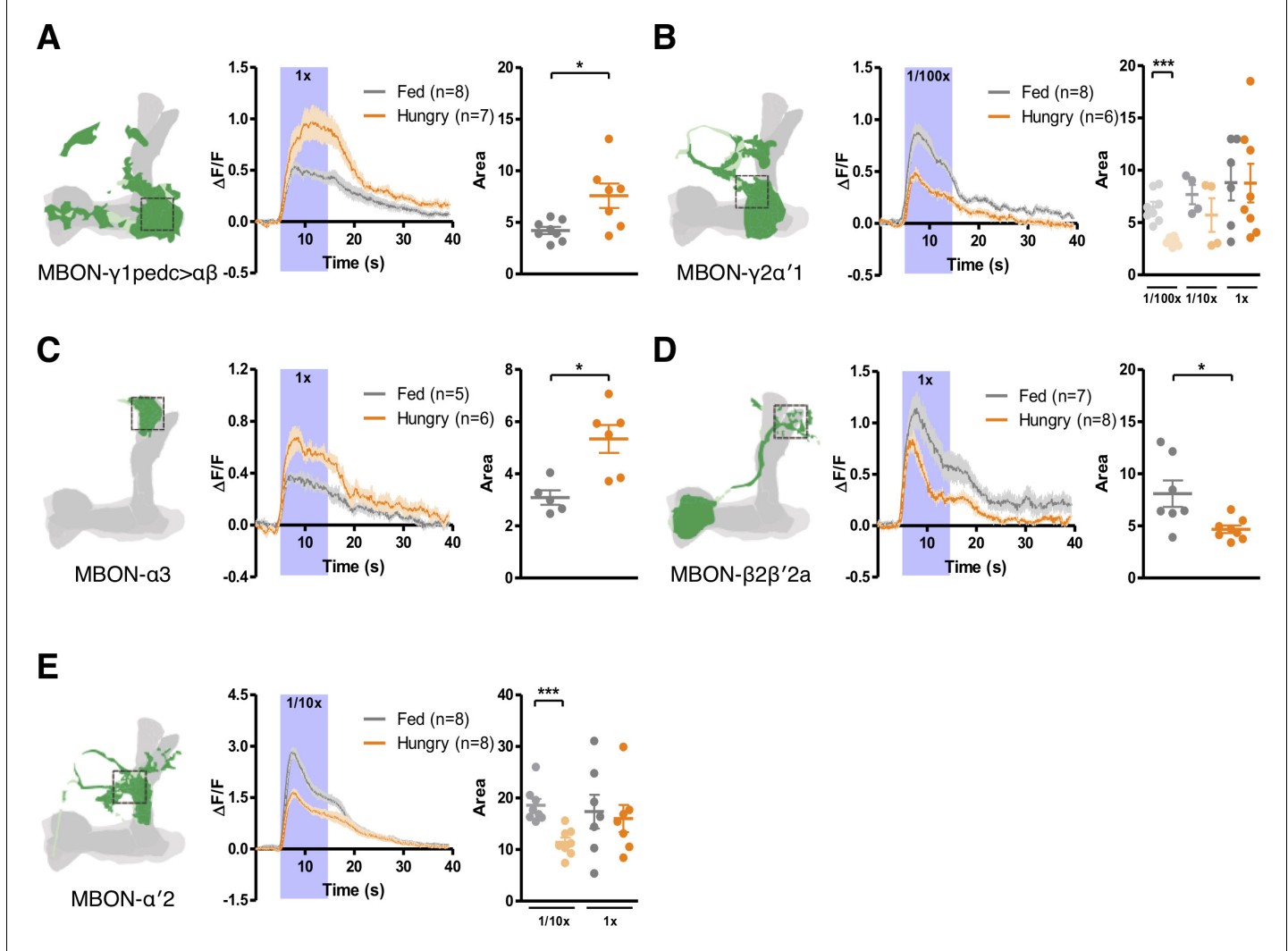

**Figure 5.** Starvation bi-directionally modulates the responses of MBONs to yeast odor. Hunger increases (**A and C**) and decreases (**B, D and E**) yeast odor-evoked calcium transients (visualized using GCaM6m) in (**A**) MBON-γ1pedc>αβ (with *MB112C-splitGAL4*), (**B**) MBON-γ2α'1 (with *MB077B-splitGAL4*), (**C**) MBON-α3 (with *G0239-GAL4*), (**D**) MBON-β2β'2a (with *MB399B-splitGAL4*), and (**E**) MBON-α'2 (with *MB091C-splitGAL4*). Schematics indicate where the Ca²⁺ response was measured. Ca²⁺ imaging data are mean (solid line) ± SEM (shaded area) normalized curves (see Materials and methods). Wide purple bars indicate the 10 s when yeast odor was presented. Dot plots are quantifications of the area under the curve during the 10 s odor presentation. Individual data points and mean ± SEM are shown. Yeast odor was also tested at 1/10x dilution in (**B**) and (**E**) and at 1/100x dilution in (**B**). Asterisks denote statistical significance; Mann-Whitney test; (**A**) p=0.0289, (**B**) p=0.0007, (**C**) p=0.0173, (**D**) p=0.0289, (**E**) p=0.0003.

DOI: https://doi.org/10.7554/eLife.35264.014

The following figure supplement is available for figure 5:

**Figure supplement 1.** Starvation does not change the responses of KCs to yeast odor.

DOI: https://doi.org/10.7554/eLife.35264.015

before testing flies in the food-seeking assay, but fly olfactory conditioning has been demonstrated to be strongest if unconditioned stimuli (in our case, the activation or silencing of DANs) come in slightly later than conditioned stimuli (odors) (*Tully and Quinn, 1985*). It is not clear whether pairing odors with activation of PAM-β2β'2a and PAM-β'2a DANs induces an aversive or appetitive memory. Nevertheless, we found that pre-conditioning yeast odor with activation or silencing of PAM-β2β'2a and PAM-β'2a DANs did not change flies' yeast food-seeking performance (*Figure 7—figure supplement 2C–F*). Finally, pairing yeast odor with silencing of MBON-γ1pedc>αβ before testing also did not influence flies' yeast food-seeking behavior *Figure 7—figure supplement 2G*, even though silencing of MBON-γ1pedc>αβ has been shown to induce an aversive memory when it is paired with

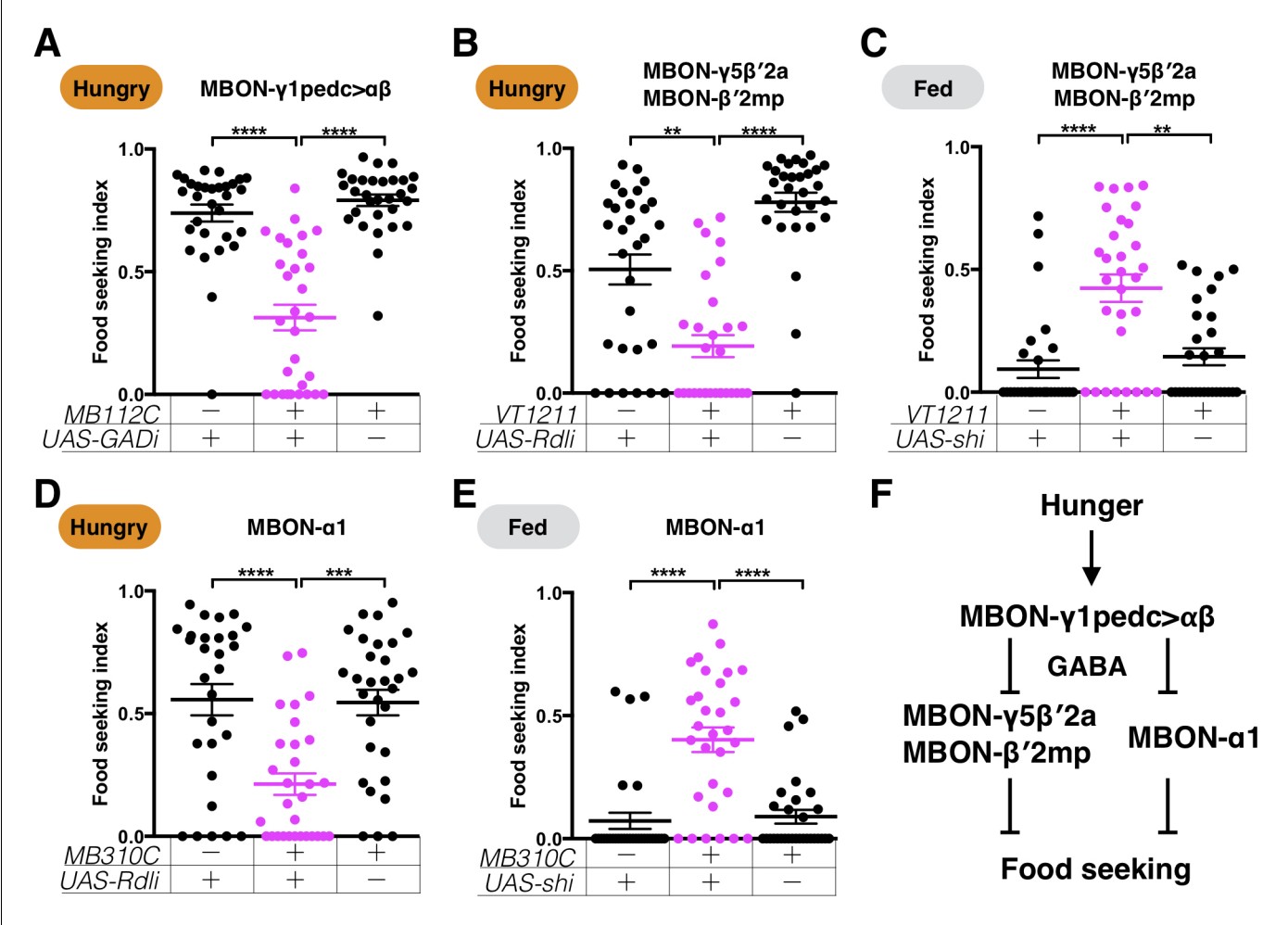

**Figure 6.** GABAergic MBON-γ1pedc>αβ promotes yeast food-seeking behavior by inhibiting β'2-innervating MBONs and MBON-α1. Male flies starved for 24 hr (**A, B and D**) or food-satiated (**C and E**) were assessed for their yeast food-seeking performance. Individual data points and mean ± SEM are shown. (**A**) The performance of *MB112C;UAS-GAD-RNAi* flies was statistically lower than the controls (Kruskal-Wallis, n = 30, p<0.0001). (**B**) The performance of *VT1211-GAL4;UAS-Rdl-RNAi* flies was significantly lower than the controls (Kruskal-Wallis, n = 30, p=0.0053). (**C**) The performance of *VT1211-GAL4;UAS-shi^{ts1}* flies was statistically higher than the controls at a restrictive 32°C (Kruskal-Wallis, n = 30, p=0.0015). (**D**) The performance of *MB310C;UAS-Rdl-RNAi* flies was statistically lower than the controls (Kruskal-Wallis, n = 28–30, p=0.0004). (**E**) The performance of *MB310C;UAS-shi^{ts1}* flies was higher than the controls at a restrictive 32°C (Kruskal-Wallis, n = 30, p<0.0021). (**F**) A model showing the relationship between MBON-γ1pedc>αβ, MBON-γ5β'2a, MBON-β'2mp, and MBON-α1.

DOI: https://doi.org/10.7554/eLife.35264.016

The following figure supplements are available for figure 6:

**Figure supplement 1.** A second set of RNAi lines confirms that GABAergic MBON-γ1pedc>αβ promotes yeast food-seeking behavior by inhibiting β'2-innervating MBONs and MBON-α1.

DOI: https://doi.org/10.7554/eLife.35264.017

**Figure supplement 2.** Expression of *UAS-shi^{ts1}* in the MBONs does not affect yeast food-seeking performance at the permissive temperature.

DOI: https://doi.org/10.7554/eLife.35264.018

**Figure supplement 3.** Knockdown efficiency of the RNAi lines.

DOI: https://doi.org/10.7554/eLife.35264.019

an odor (*Ueoka et al., 2017*). These results and the anatomy of the MB circuits indicate that the six DANs we have identified might be the main switches the hunger and satiety signals used to tune the innate behavioral response of fruit flies to the smell of food.

## The dopamine receptor DAMB functions pre- and post-synaptically to mediate hunger control

To gain more mechanistic insights into how the DANs mediate hunger control, we sought dopamine receptors that are involved in the process. We assessed yeast food-seeking behavior in hungry flies that lack each of the four dopamine receptors encoded in the fly genome (*Gotzes et al., 1994*; *Han et al., 1996*; *Hearn et al., 2002*; *Srivastava et al., 2005*). The *DAMB* and *Dop2R* mutant flies exhibited strong yeast food-seeking defects, whereas the *DopR1* and *DopEcR* mutant flies performed normally (*Figure 8A*). RNAi knockdown of *Dop2R* in the KCs and in the five food-seeking MBONs failed to recapitulate the mutant phenotype (data not shown). We speculate that *Dop2R* might be required outside the MB pathways we have identified thus far. In stark contrast, knockdown of *DAMB* in all the KCs with two independent RNAi lines and *MB010B-splitGAL4* severely compromised yeast food-seeking behavior in hungry flies (*Figure 8B* and *Figure 8*-figure supplement 1A), suggesting that the DANs regulate the response of flies to yeast food by pre-synaptically modulating KC-to-MBON connectivity via the DAMB receptor. This is not surprising because DAMB is strongly expressed in all KC types (*Han et al., 1996*). However, we found that DAMB in the MBONs is also required for yeast food-seeking behavior. Knockdown of DAMB in MBON-β2β′2a, MBON-γ2α′1, and MBON-α3 with two independent RNAi lines significantly impaired yeast food-seeking behavior in hungry flies (*Figure 8C–E* and *Figure 8—figure supplement 1B–D*). Furthermore, knockdown of DAMB in MBON-1pedc>αβ with two independent RNAi lines promoted yeast food-seeking behavior in fed flies (*Figure 8F* and *Figure 8—figure supplement 1E*). DAMB knockdown in MBON-α′2 had no detectable effect on yeast food-seeking behavior in hungry flies (*Figure 8G* and *Figure 8—figure supplement 1F*). Taken together, these data suggest that the DANs modulate the MB circuit both pre- and post-synaptically via the DAMB receptor to control the behavioral response of flies to yeast food.

## Physiological properties of the DANs are modulated by starvation

Our genetics and behavioral results suggest that PPL1-α3, PAM-β2β′2a, PAM-β′2a, PPL1-α′2α2, and PPL1-γ2α′1 DANs are positively regulated by hunger, while PPL1-γ1pedc DANs are suppressed by hunger. To search for evidence that these DANs are modulated by starvation, we first checked their spontaneous activities using in vivo functional imaging according to previous studies (*Cervantes-Sandoval et al., 2017*; *Plaçais et al., 2012*). However, we found the spontaneous activities to be variable among individuals and we failed to identify obvious differences between fed and hungry flies. We then checked the odor-evoked responses in these DANs. Some DANs have been shown to respond to odors (*Cervantes-Sandoval et al., 2017*; *Felsenberg et al., 2017*), and we reasoned that changes in the odor-evoked responses of these DANs could be detected if their excitability is modulated by hunger. Interestingly, we found that starvation decreased the yeast odor-evoked calcium transient in PPL1-γ1pedc DANs but increased it in PPL1-α3 DANs, consistent with our behavioral data (*Figure 9A and B*). However, we did not detect starvation-induced change in odor-evoked responses of PAM-β2β′2a and PPL1-α′2α2 DANs (*Figure 9C and D*). Also, PAM-β′2a and PPL1-γ2α′1 DANs showed small and inconsistent responses to yeast odor in both hungry and fed flies (data not shown). Recent studies have found that some DANs receive direct inputs from the KCs (*Cervantes-Sandoval et al., 2017*; *Takemura et al., 2017*). It is noteworthy that PPL1-γ1pedc and MBON-γ 1pedc>αβ innervate the same compartment in the MB lobes, but their yeast odor-evoked responses are contrastingly modulated by starvation (*Figures 5A* and *9A*). This supports our notion that the starvation-induced modulations we observed are not due to changes in circuits upstream of the KCs. The DANs may also receive other unidentified olfactory inputs, so although our data suggest that PPL1-γ1pedc and PPL1-α3 DANs are modulated by starvation, we acknowledge that it remains possible that the modulations happen in other parts of the input pathways.

A recent study showed that protein starvation changes the distribution of active zones in a specific type of DANs innervating the wedge neuropil (*Liu et al., 2017*). Therefore, we expressed an active zone marker DSyd-1-GFP (*Owald et al., 2010*) in our identified DANs and found that the mean intensity of DSyd-1-GFP in PPL1-γ2α′1, PAM-β2β′2a, and PPL1-α′2α2 DANs is statistically higher in hungry flies (*Figure 9—figure supplement 1A–C*). A UAS-DenMark transgene (*Nicolaï et al., 2010*) was also co-expressed in these DANs. Although DenMark is a dendritic marker (*Nicolaï et al., 2010*), we readily detected its signal in the axonal processes of PPL1-γ2α′1, PAM-β

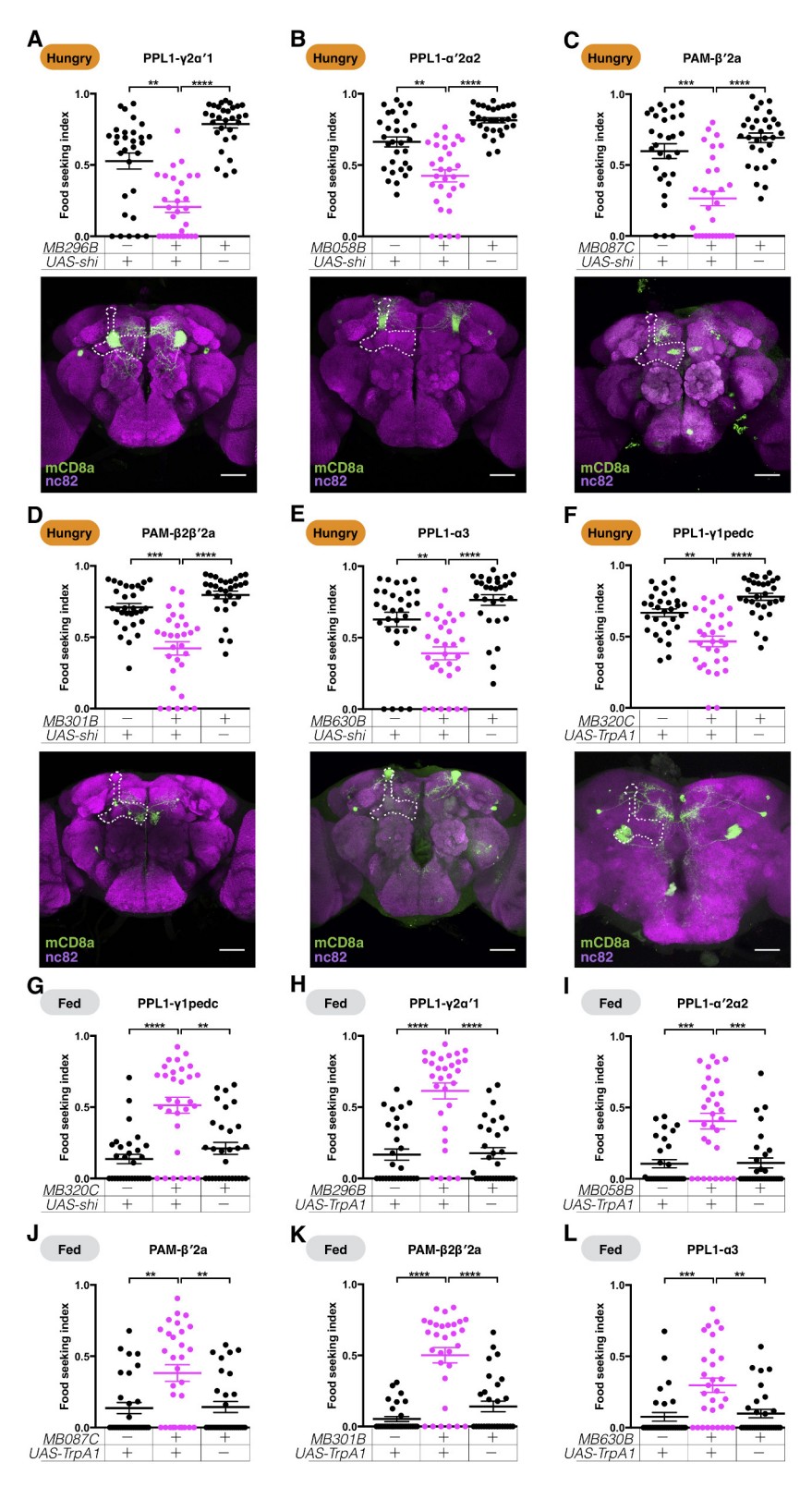

**Figure 7.** DANs mediate hunger-control of yeast food-seeking behavior. (**A–E**) Male flies starved for 24 hr were assessed for their yeast food-seeking performance. At a restrictive 32°C, the performance was significantly different between the controls and flies expressing *UAS-shi^{ts1}* in (**A**) PPL1-γ2α'1 (*MB296B,* Kruskal-Wallis, n = 30, p=0.0018), (**B**) PPL1-α'2α2 (*MB058B,* Kruskal-Wallis, n = 30, p=0.0012), (**C**) PAM-β'2a (*MB087C,* Kruskal-Wallis, n = 30, p=0.0002), (**D**) PAM-β2β'2a (*MB301B,* Kruskal-Wallis, n = 30, p=0.0001), and (**E**) PPL1-α3 (*MB630B,* Kruskal-Wallis, n = 30, p=0.0023) DANs. (**F**) The
*Figure 7 continued on next page*

*Figure 7 continued*

performance of *MB320C;UAS-TrpA1* male flies starved for 24 hr was lower than the controls (PPL1-γ1pedc, Kruskal-Wallis, n = 30, p=0.003). (G) The performance of male *MB320C;UAS- shi^ts1* fed flies was statistically better than the controls (PPL1-γ1pedc, Kruskal-Wallis, n = 30, p=0.001). (H–L) Food-satiated male flies were tested for their yeast food-seeking performance. At 32°C, the performance was statistically different between the controls and flies expressing *UAS-TrpA1* in (H) PPL1-γ2α′1 (*MB296B*, Kruskal-Wallis, n = 30, p<0.0001), (I) PPL1-α′2α2 (*MB058B*, n = 30, p=0.0004), (J) PAM-β′2a (*MB087C*, Kruskal-Wallis, n = 30, p=0.0056), (K) PAM-β2β′2a (*MB301B*, Kruskal-Wallis, n = 30, p<0.0001), and (L) PPL1-α3 (*MB630B*, Kruskal-Wallis, n = 30, p=0.0049) DANs. Individual data points and mean ± SEM are shown. The brain images in (A–F) are full z-projections of confocal stacks showing the expression patterns of the GAL4 lines (green) counter-stained with nc82 antibody (magenta). One side of the MB is outlined by a white dashed line. Scale bars are 100 μm.

DOI: https://doi.org/10.7554/eLife.35264.020

The following figure supplements are available for figure 7:

**Figure supplement 1.** Expression of *UAS-shi^ts1* or *UAS-TrpA1* in the DANs does not affect yeast food-seeking performance at the permissive temperature.

DOI: https://doi.org/10.7554/eLife.35264.021

**Figure supplement 2.** Pre-conditioning flies by pairing yeast odor with the activation or silencing of DANs and MBONs does not affect their yeast food-seeking performance.

DOI: https://doi.org/10.7554/eLife.35264.022

2β′2a, and PPL1-α′2α2 DANs. Importantly, the mean intensity of the DenMark signals in these DANs did not differ between fed and hungry flies (*Figure 9—figure supplement 1A–C*), suggesting that the increase of DSyd-1-GFP signal is not due to higher GAL4 activity or other non-specific effects caused by starvation. These results indicate that starvation may increase the density or size of the active zones in PPL1-γ2α′1, PAM-β2β′2a, and PPL1-α′2α2 DANs, elevating their dopamine release in hungry flies. However, we note that further experiments are necessary to compare endogenous active zone proteins between fed and starved flies in order to fully support this conclusion. We did not check DSyd-1-GFP signal in PAM-β′2a DANs, because the flies failed to survive to adulthood when the expression of DSyd-1-GFP was driven by *MB087C-splitGAL4*, that is the line we used to label PAM-β′2a DANs. Moreover, we did not detect starvation-induced change in DSyd-1-GFP signal in PPL1-γ1pedc or PPL1-α3 DANs (*Figure 9—figure supplement 1D and E*). Taken together, our data provide evidence for potential complex and cell-type-specific hunger modulations in the DANs that control food-seeking behavior. Nevertheless, further comprehensive studies are needed to provide a complete picture of how hunger modulates the physiological properties of these DANs.

## DANs are differentially regulated by hunger and satiety signals

We have identified six types of DANs as the main switches regulated by starvation to promote food-seeking behavior. We next examined the potential hunger and satiety inputs that these DANs might receive. A recent study demonstrated that serotonin is a general hunger signal that affects a wide range of feeding-related behaviors in the fly (*Albin et al., 2015*). To examine whether serotonin also regulates hunger-evoked yeast food-seeking behavior, we first used *R50H05-GAL4* (*Albin et al., 2015*) and *UAS-shi^ts1* to block serotonergic neurons in hungry flies. We observed a strong yeast food-seeking defect in these flies at a restrictive 32°C, but not at a permissive 23°C (*Figure 10A* and *Figure 10—figure supplement 1A*). Conversely, artificial activation of the *R50H05*-positive serotoninergic neurons with *UAS-TrpA1* promoted yeast food-seeking behavior in fed flies at 32°C, but not at 23°C (*Figure 10B* and *Figure 10—figure supplement 1B*). We next examined whether the effect of serotonin is mediated by the six food-seeking DANs through RNAi knockdown of each of the five serotonin receptors in these DANs (*Figure 10C–F* and *Figure 10—figure supplement 2*). We found that serotonin receptors in PPL1-γ2α′1 and PPL1-γ1pedc DANs are critical for yeast food-seeking behavior in hungry flies (*Figure 10C–F*), suggesting that hunger-evoked serotonin release may control the response of flies to the smell of yeast food through these two DANs. Interestingly, the two DAN types are regulated by different serotonin receptors. Knockdown of 5HT1B receptors in PPL1-γ2α′1 DANs with two independent RNAi lines impaired yeast food-seeking behavior (*Figure 10C*, *Figure 10—figure supplement 3A*), but no significant defect was detected when the same receptor was knocked down in PPL1-γ1pedc DANs (*Figure 10D*). In contrast, no phenotypic deficiency was observed when 5HT2A receptors were knocked down in PPL-γ2α′1 DANs (*Figure 10E*), whereas knockdown of 5HT2A receptors with two independent RNAi lines in the PPL1-γ1pedc DANs strongly

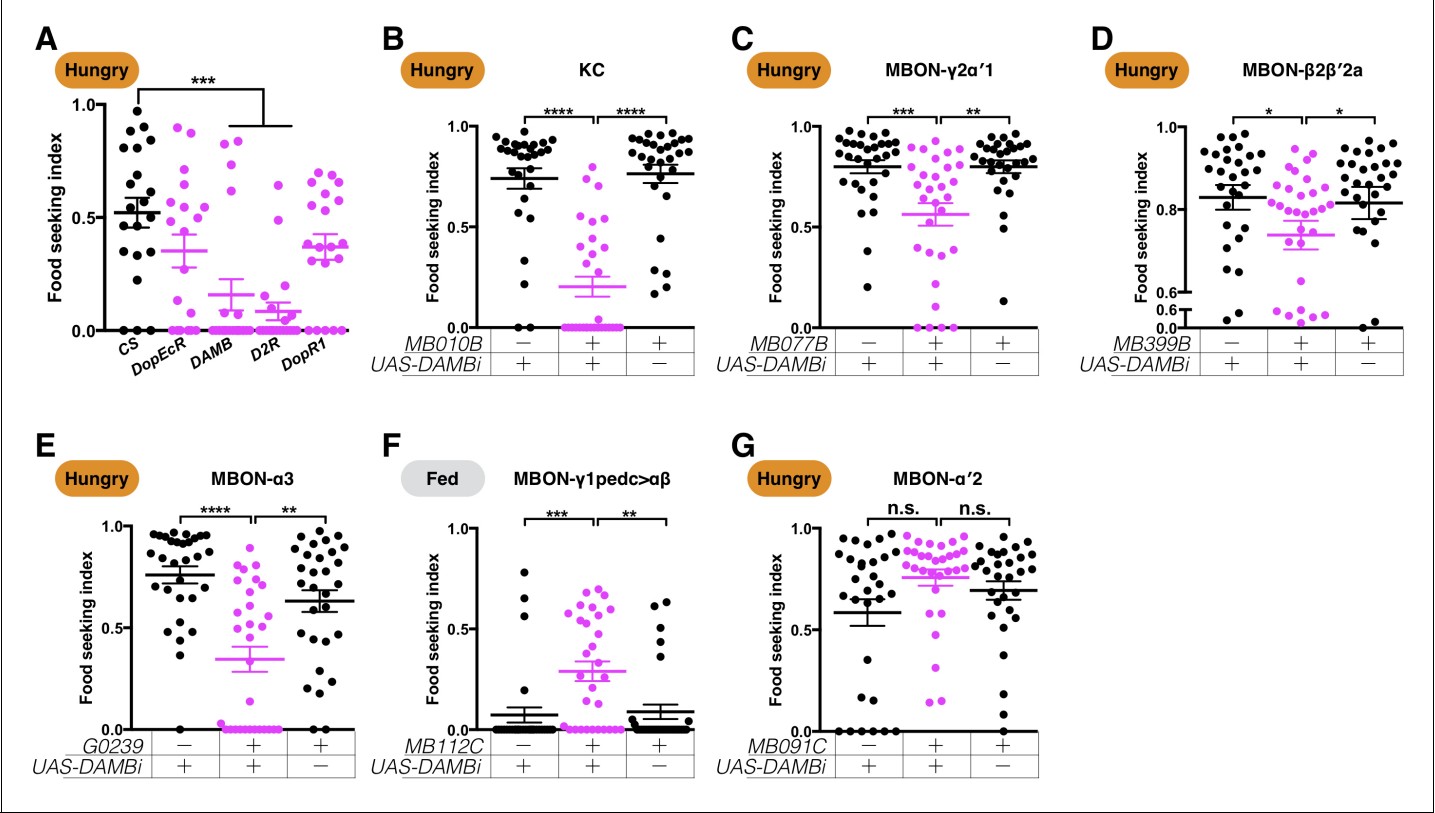

**Figure 8.** The dopamine receptor DAMB is required pre- and post-synaptically to regulate yeast food-seeking behavior. (**A**) The yeast food-seeking performance in 24-hr-starved male wild-type flies (CS, n = 21) and flies homozygous for *DopEcR* (n = 19), *DAMB* (n = 20), *D2R* (n = 20), and *DopR1* (n = 20) was assessed. The performance of the *DAMB* and *D2R* flies was significantly lower than for the wild-type flies (Kruskal-Wallis, p=0.0024 for *DAMB*; *p*=0.0003 for *D2R*). (**B–G**) Male flies starved for 24 hr (**B–E** and **G**) or food-satiated (**F**) were assessed for their yeast food-seeking performance. The performance of *GAL4;UAS-DAMB-RNAi* flies was statistically different from the controls for (**B**) *MB010B* (all KCs, Kruskal-Wallis, n = 29–30, p<0.0001), (**C**) *MB077B* (MBON-γ2α′1, Kruskal-Wallis, n = 30, p=0.0018), (**D**) *MB399B* (MBON-β2β′2a, Kruskal-Wallis, n = 29–30, p=0.0285), (**E**) *G0239* (MBON-α3, Kruskal-Wallis, n = 30, p=0.007). (**F**) *MB112C* (MBON-γ1pedc>αβ, Kruskal-Wallis, n = 30, p=0.0028), but not for (**G**) *MB091C* (MBON-α′2, Kruskal-Wallis, n = 30, p=0.5757). Satiety states (fed or hungry) are indicated in each figure. Individual data points and mean ± SEM are shown.
DOI: https://doi.org/10.7554/eLife.35264.023

The following figure supplement is available for figure 8:

**Figure supplement 1.** A second set of RNAi lines confirms that the dopamine receptor DAMB is required both pre- and post-synaptically to regulate yeast food-seeking behavior.
DOI: https://doi.org/10.7554/eLife.35264.024

impaired yeast food-seeking behavior (*Figure 10F* and *Figure 10—figure supplement 3B*). Differential usage of these serotonin receptors may potentially explain how starvation activates PPL1-γ2α′1 DANs but inhibits PPL1-γ1pedc DANs. In support of this idea, different 5HT receptors have been shown to have opposing effects on cAMP signaling when expressed in mammalian cells (*Saudou et al., 1992*).

Next, we examined whether the DANs receive other hunger signals. NPF and sNPF are hunger-evoked neuropeptides, and NPF is suggested to inhibit the activity of PPL1-γ1pedc when flies are hungry (*Krashes et al., 2009*). Consistently, when NPF receptors (NPFR) were knocked down with two independent RNAi lines in PPL1-γ1pedc DANs, the yeast-seeking performances of hungry flies diminished significantly (*Figure 11A* and *Figure 11—figure supplement 1A*). In addition, we found that knockdown of sNPF receptors (sNPFR) with two independent RNAi lines in PPL1-γ1pedc DANs impaired yeast food-seeking behavior (*Figure 11B* and *Figure 11—figure supplement 1B*), suggesting that serotonin, NPF, and sNPF work together to suppress PPL1-γ1pedc DANs in hungry flies. We then tested whether NPFR and sNPFR are required in other yeast-seeking DANs. Knockdown of NPFR in PAM-β′2a, PPL1-α3 and PAM-β2β′2a DANs, but not the other two DAN types, impaired

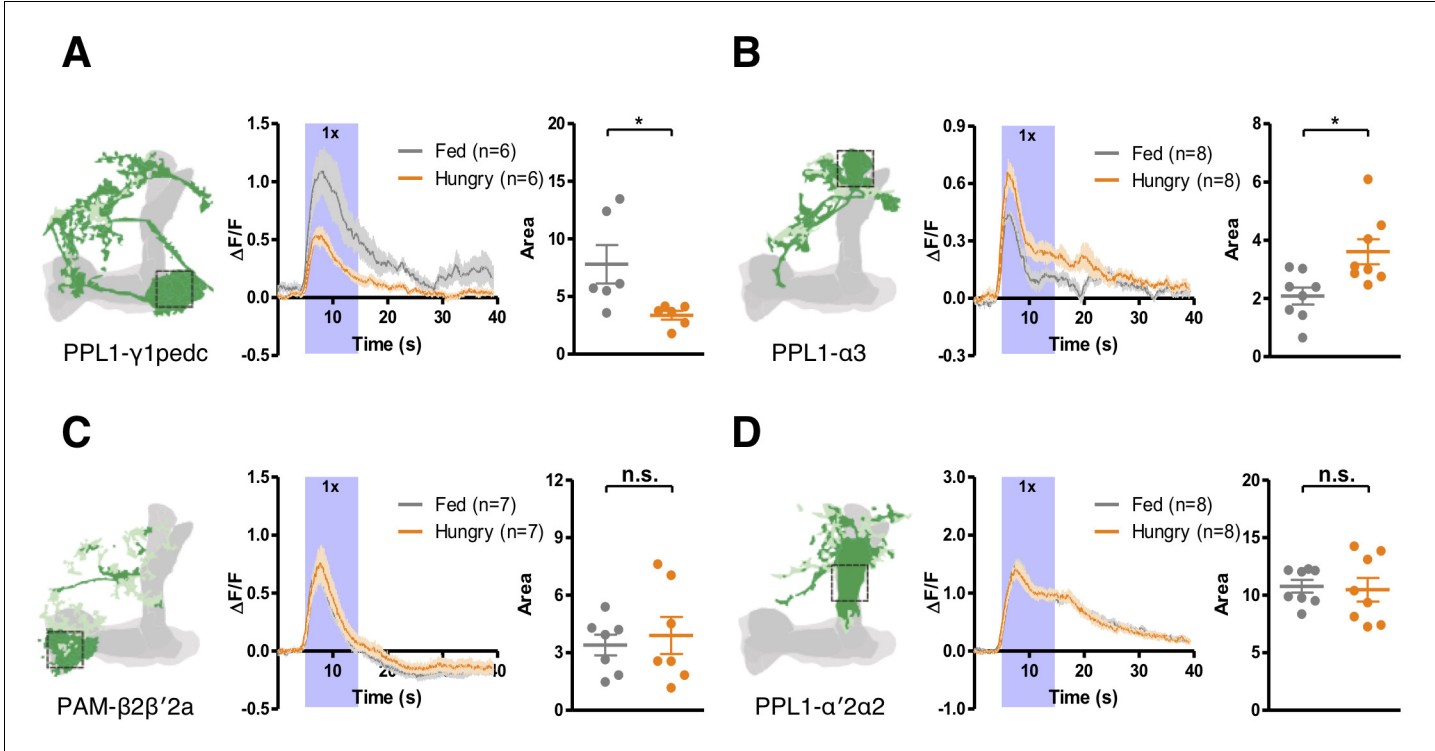

**Figure 9.** Starvation modulates yeast odor-evoked responses in some DANs. GCaMP6m was expressed in PPL1-γ1pedc DANs using *MB320C split-GAL4* (A), PPL1-α3 DANs using *G0239-GAL4* (B), PAM-β2β'2a DANs using *MB301B split-GAL4* (C), and PPL1-α'2α2 DANs using *MB058B split-GAL4* (D). The $Ca^{2+}$ signals were measured when flies were presented with yeast odor. Schematics indicate where the $Ca^{2+}$ response was measured. $Ca^{2+}$ imaging data are mean (solid line) ± SEM (shaded area) normalized curves (see Materials and methods). Wide purple bars indicate the 10 s when yeast odor was presented. Dot plots are quantifications of the area under the curve during the 10 s odor presentation. Individual data points and mean ± SEM are shown. Statistical differences were detected in (A) PPL1-γ1pedc (Mann-Whitney test, n = 6, p=0.026) and (B) PPL1-α3 (Mann-Whitney test, n = 8, p=0.014) DANs, but not in (C) PAM-β2β'2a (Mann-Whitney test, n = 7–8, p=0.7789) or (D) PPL1-α'2α2 (Mann-Whitney test, n = 8, p=0.7984) DANs.

DOI: https://doi.org/10.7554/eLife.35264.025

The following figure supplement is available for figure 9:

**Figure supplement 1.** Starvation may increase the density or size of active zones in some DANs.

DOI: https://doi.org/10.7554/eLife.35264.026

yeast food-seeking behavior in hungry flies (*Figure 11C,E,G,I and K*; *Figure 11—figure supplement 1H,K and M*). In contrast, knockdown of sNPFR in all but PPL1-α'2α2 DANs reduced the yeast food-seeking behavior of hungry flies (*Figure 11D,F,H,J and L*; *Figure 11—figure supplement 1C,I,L and N*). These results argue that these DANs are regulated by multiple hunger signals in a cell-type-specific manner.

In addition to the hunger signals, we assessed insulin-like peptides (ILPs) and Allatostatin A (AstA) signaling pathways that both mediate satiety in the fly (*Hergarden et al., 2012*; *Root et al., 2011*; *Yu et al., 2016*). RNAi knockdown of *Insulin receptor* (*dInR*) in PAM-β'2a, PPL1-α'2α2, and PPL1-γ2α'1 DANs promoted yeast food-seeking behavior in fed flies (*Figure 11O,Q and S*; *Figure 11—figure supplement 1D,F and J*), whereas no altered phenotype was observed when *dInR* was knocked down in PPL1-α3, PAM-β2β'2a, and PPL1-γ1pedc DANs (*Figure 11M,U and W*). We performed similar experiments for the AstA receptor (DAR1) (*Lenz et al., 2000*). Flies expressing two independent *DAR1* RNAi (*Yamagata et al., 2016*) in PPL1-γ2α'1, PPL1-α'2α2, and PAM-β2β'2a DANs exhibited increased food-seeking behavior when they were food-satiated (*Figure 11P,R and X*; *Figure 11—figure supplement 1E,G and O*). No phenotypic difference was detected when *DAR1* was knocked down in PPL1-α3, PAM-β'2a, and PPL1-γ1pedc DANs (*Figure 11N,T and V*). Overall, our RNAi knockdown data suggest potential rich hunger and satiety inputs to the MB circuits that control the innate behavioral response of flies to the smell of yeast food (*Figure 12*).

## Discussion

### The MB regulates innate olfactory behavior

The MB has been extensively studied for its role in olfactory associative learning and memory (*Davis, 2005*; *Heisenberg, 2003*; *Keene and Waddell, 2007*) and was initially considered to be dispensable for innate olfactory behavior (*de Belle and Heisenberg, 1994*). The random connectivity between antennal lobe projection neurons and the KCs makes the MB an ideal circuit for olfactory learning, but not for encoding innate odor valences (*Caron et al., 2013*). However, recent studies suggest that the MB also plays a role in innate odor responses. Switching off KC outputs impairs the responses of flies to attractive odors at low concentration (*Wang et al., 2003*) and decreases their aversion to $CO_2$ when they are hungry (*Bräcker et al., 2013*). Blocking the neurotransmission of β′2-innervating MBONs reduces the avoidance behavior of flies to aversive odors (*Owald et al., 2015*; *Perisse et al., 2016*). Here, we provide further evidence to support the role of MB circuits in innate olfactory behavior. Switching off the output synapses of each of the three major KC types compromises the abilities of flies to seek food odors. Five MB output neurons—MBON-α3, MBON-β2β′2a, MBON-α′2, MBON-γ2α′1, and MBON-γ1pedc>αβ—collectively drive hungry flies to follow the smell of food. MBON-γ1pedc>αβ does so by inhibiting MBON-α1, MBON-β′2mp and MBON-γ5β′2a. Our data, together with previous studies, suggest that the MB is a value-processing center for both trained and untrained odors.

### The MB potentially receives rich inputs of hunger and satiety signals

An unexpected finding of our study is that the MB may receive remarkably rich inputs of hunger and satiety signals. It has previously been shown that PPL1-γ1pedc DANs are regulated by NPF (*Krashes et al., 2009*). Here, we tested two satiety signals (insulin and AstA) and three hunger signals (serotonin, sNPF, and NPF), and found that the six DAN types regulating food-seeking behavior are differentially modulated by combinations of these signals. PPL1-γ1pedc and PPL1-α3 DANs are regulated exclusively by hunger signals, whereas PPL1-α′2α2 DANs only receive satiety signals. The other three DAN types are regulated by both signals. Our survey is by no means complete, so hunger and satiety regulation of these DANs might be even richer. It is important to point out that although our RNAi experiments suggest that the food-seeking DANs receive direct inputs of hunger and satiety signals, further study is required to identify their upstream serotonergic and peptidergic neurons, showing the functional connections to these DANs and demonstrating the expression and cellular localization of the receptors for hunger and satiety signals in them. If these DANs indeed receive direct hunger and satiety input signals, it would be interesting to understand the purpose of such rich regulation and how a single DAN integrates multiple hunger and satiety signals. Nevertheless, our data imply that the satiety state could have a strong impact on MB circuit computations. Indeed, in addition to controlling appetitive memory expression (*Krashes et al., 2009*; *Perisse et al., 2016*), hunger also gates memory formation (*Hirano et al., 2013*; *Huetteroth et al., 2015*; *Musso et al., 2015*; *Plaçais et al., 2017*). In particular, feeding flies nutritious sugar after olfactory learning changes the activity of PPL1-γ1pedc DANs and this change upregulates energy metabolism in the MB to promote long-term memory consolidation (*Musso et al., 2015*; *Plaçais et al., 2017*). Nutritious sugar feeding has also been shown to immediately suppress the PAM-γ3 DANs via the AstA signaling pathway, and this suppression provides a positive reinforcing signal for the formation of sugar-odor associative memory (*Yamagata et al., 2016*). Furthermore, starvation recruits the MB for $CO_2$ avoidance behavior (*Bräcker et al., 2013*). The MB circuit has been shown to be involved in thirst-driven water-seeking (*Lin et al., 2014b*), sleep (*Joiner et al., 2006*; *Pitman et al., 2006*), decision-making (*DasGupta et al., 2014*), and temperature preference behavior (*Bang et al., 2011*; *Hong et al., 2008*; *Shih et al., 2015*). Given the broad influence of hunger and satiety on the MB-innervating DANs, the MB may serve as an integration center for hunger to interact with other motivations, giving rise to hierarchical behaviors.

### Hunger tunes the response of MBONs to yeast odor

Our study reveals that hunger tunes the response of MBONs to yeast odor in a cell-type-specific manner. The outputs of the five MBONs we have identified are required for food-seeking behavior. Since flies only seek food when they are hungry, it could be expected that starvation potentiates

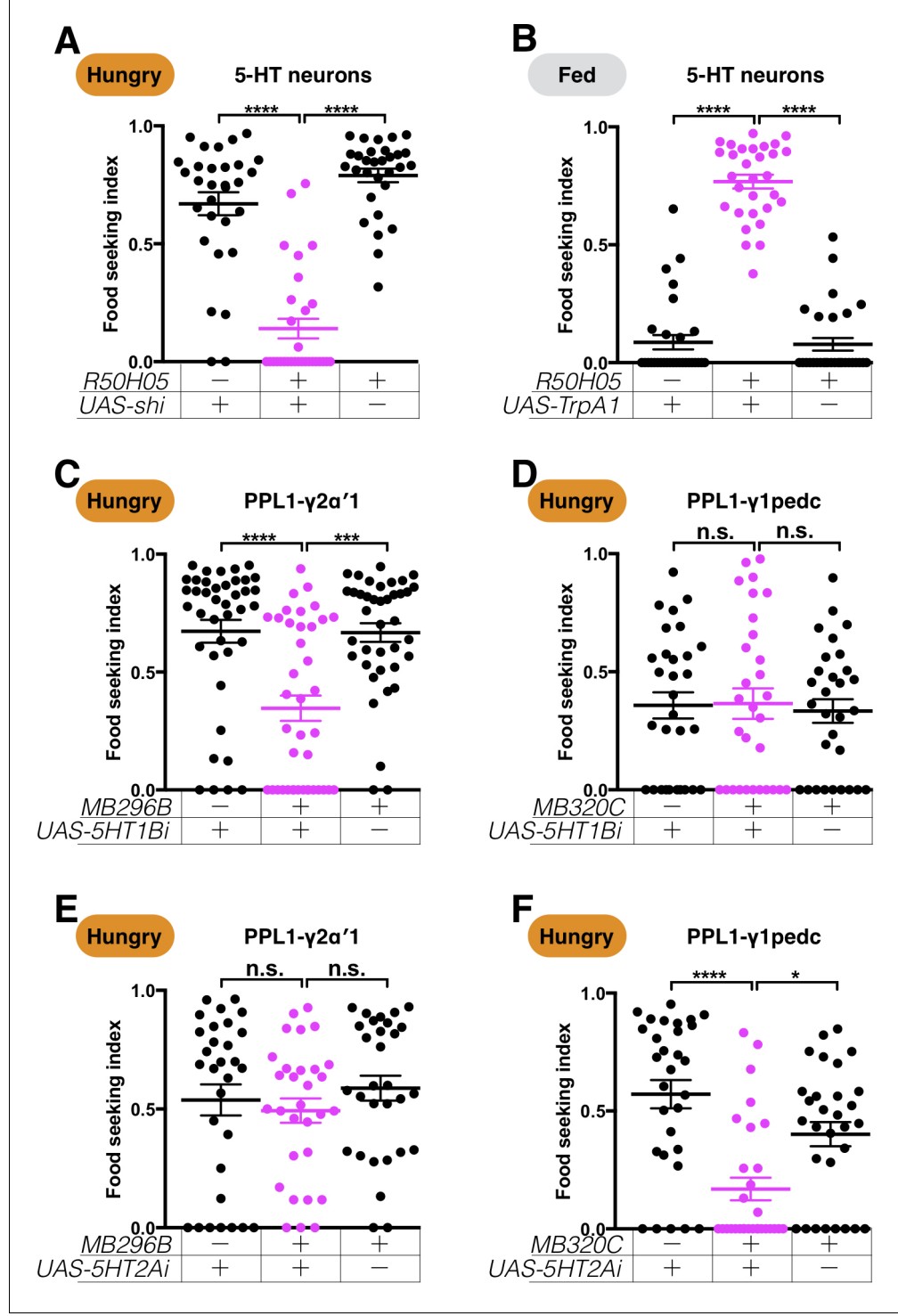

**Figure 10.** Serotonin regulates PPL1-γ2α′1 and PPL1-γ1pedc DANs via different receptors. The yeast food-seeking performance of 24-hr-starved (**A and C–F**) and food-satiated (**B**) male flies was assessed. (**A**) The performance of *R50H05-GAL4;UAS-shi^{ts1}* flies was statistically worse than for the control flies at a restrictive 32°C (5-HT neurons, Kruskal-Wallis, n = 30, p<0.0001). (**B**) The performance of *R50H05-GAL4;UAS-TrpA1* flies was statistically better than for the control flies at a restrictive 32°C (5-HT neurons, Kruskal-Wallis, n = 30, p<0.0001). (**C**) The performance of *MB296B;UAS-5HT1B-RNAi* flies was statistically worse than for the control flies (PPL1-γ2α′1, Kruskal-Wallis, n = 39–40, p=0.0003). (**D**) The performance of *MB320C;UAS-5HT1B-RNAi* flies was not statistically different from that of control flies (PPL1-γ1pedc, Kruskal-Wallis, n = 30, p>0.9999). (**E**) The performance of *MB296B;UAS-5HT2A-*

*Figure 10 continued on next page*

*Figure 10 continued*

*RNAi* flies was not statistically different from that of control flies (PPL1-γ2α′1 Kruskal-Wallis, n = 30, p=0.5116). (**F**) The performance of *MB320C;UAS-5HT2A-RNAi* flies was statistically worse than that of control flies (PPL1-γ1pedc, Kruskal-Wallis, n = 30, p=0.0263). Individual data points and mean ± SEM are shown.

DOI: https://doi.org/10.7554/eLife.35264.027

The following figure supplements are available for figure 10:

**Figure supplement 1.** Expression of *UAS-shi^{ts1}* or *UAS-TrpA1* in serotoninergic neurons does not affect yeast food-seeking performance at the permissive temperature.

DOI: https://doi.org/10.7554/eLife.35264.028

**Figure supplement 2.** RNAi knockdown of serotonin receptors in the yeast-seeking DANs.

DOI: https://doi.org/10.7554/eLife.35264.029

**Figure supplement 3.** A second set of RNAi lines confirms the importance of 5HT1B and 5HT2A receptors in regulating yeast food-seeking behavior.

DOI: https://doi.org/10.7554/eLife.35264.030

food odor-evoked responses in these MBONs. Indeed, starvation increases yeast-odor responses in MBON-α3 and MBON-γ1pedc>αβ; yet, intriguingly, it depresses the responses in the other three MBONs. These unexpected results suggest that the level of odor-evoked response might be critical in some MBONs. We propose a putative circuit mechanism to explain this phenomenon (*Figure 12— figure supplement 1*). In this putative configuration, MBONs like MBON-β2β′2a, MBON-α′2, and MBON-γ2α′1 connect to two downstream neurons with different connectivity strength, and the weakly connected downstream neuron (neuron B) inhibits the strongly connected one (neuron A). Under this configuration, only weakly evoked MBONs under starvation will engage neuron A and induce food-seeking behavior. Importantly, in this model, blocking the MBON will also impair food-seeking behavior in hungry flies. A similar activity level-dependent mechanism has been observed in courtship-promoting P1 interneurons in the fly (*Hoopfer et al., 2015*). Low-frequency activation of P1 neurons evokes aggression, whereas high-frequency activation of the same neurons inhibits aggression but promotes wing extension. Our hypothetical model remains to be investigated and there are certainly many other possible circuit configurations that can achieve a similar effect. It will be important to identify the neural circuits downstream of the MBONs to understand how the outputs of the MBONs are integrated and translated into food-seeking behavior.

## DAMB is critical for hunger control of food-seeking behavior

Hunger-induced modulation appears to occur both pre- and post-synaptically through DAMB receptors. Knockdown of DAMB either in KCs or MBONs (except MBON-α′2) impairs yeast food-seeking behavior in hungry flies. DAMB is highly expressed in all KCs, and immunostaining results have showed that DAMB is mainly distributed in axonal processes, but is absent from dendrites (*Han et al., 1996*). However, our data suggest that DAMB might also function in the dendrites of the MBONs. Consistent with this possibility, a recent study found that DAMB is required in MBON-α′3 to allow flies to become accustomed to novel odors (*Hattori et al., 2017*). In the same study, a GAL4 transgene inserted in the genomic locus of DAMB labels MBON-α′3. Nevertheless, it remains to be demonstrated that the MBONs required for food-seeking behavior also express DAMB and that DAMB can be localized to the dendrites of these MBONs.

Another dopamine receptor, DopR1, is also highly expressed in the KCs (*Kim et al., 2003*). DopR1 has been shown to be critical for the acquisition of olfactory memories (*Kim et al., 2007*; *Qin et al., 2012*), whereas DAMB is mainly required for memory maintenance (*Berry et al., 2012*; *Musso et al., 2015*; *Plaçais et al., 2017*). This scenario raises an interesting question as to how these two dopamine receptors are differentially utilized by the MB circuit. An attractive hypothesis is that phasic dopamine release evoked by rewards or punishments during learning activates DopR1, and tonic (or spontaneous) dopamine release mainly engages DAMB (*Berry and Davis, 2014*; *Ichinose et al., 2017*). Although this hypothesis remains to be tested, our study provides another supportive example of satiety state, which presumably modulates DAN activities in a mild and sustained manner, controlling food-seeking behavior via DAMB.

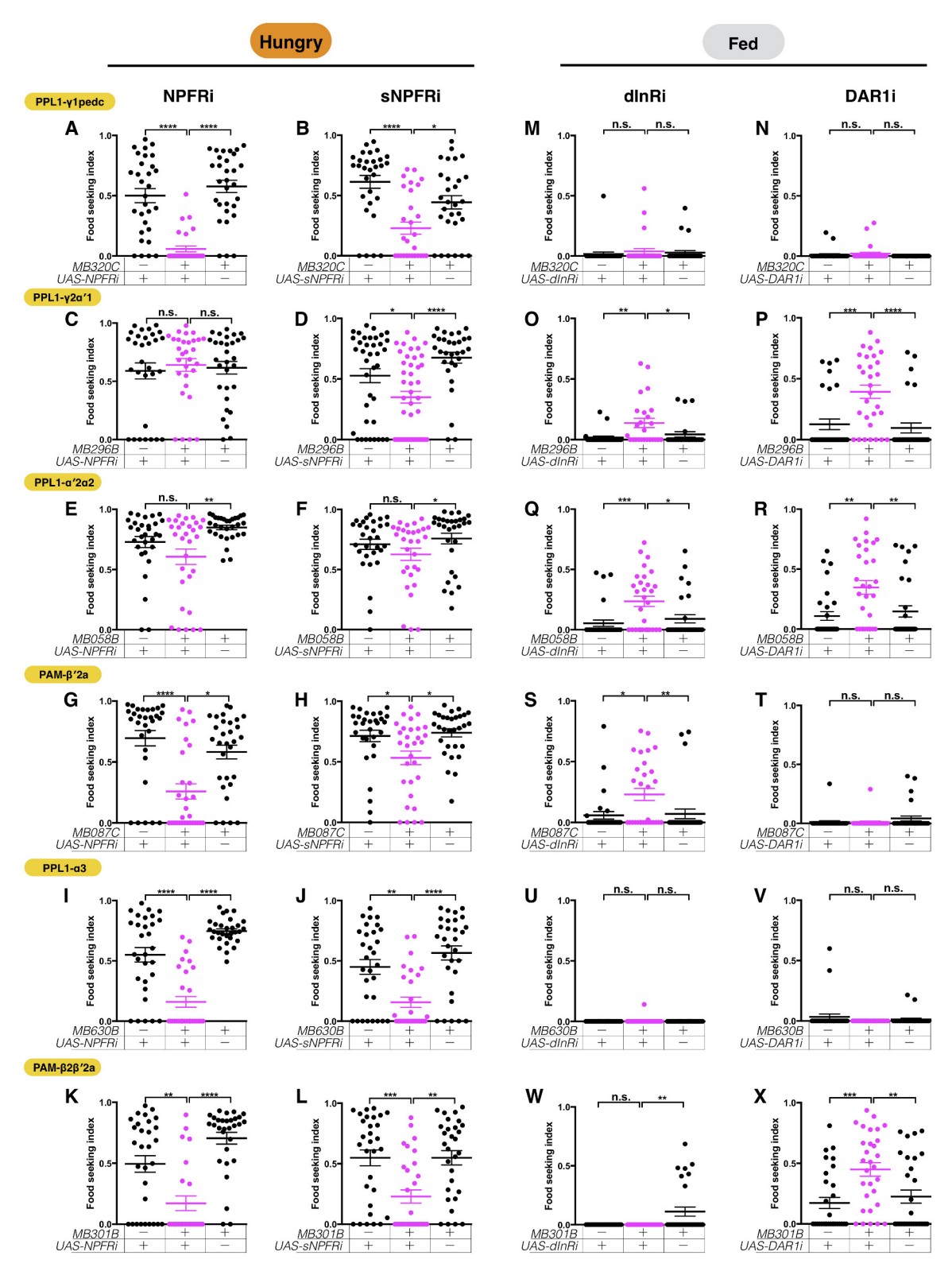

**Figure 11.** The six assessed DANs are regulated by combinations of hunger and satiety signals. The yeast food-seeking performance of 24-hr-starved (**A–L**) and food-satiated (**M–X**) male flies was assessed. (**A**) The performance of *MB320C;UAS-NPFR-RNAi* flies was impaired (PPL1-γ1pedc, Kruskal-Wallis, n = 30, p<0.0001). (**B**) The performance of *MB320C;UAS-sNPFR-RNAi* flies was impaired (PPL1-γ1pedc, Kruskal-Wallis, n = 29–30, p=0.0438). (**C**) The performance of *MB296B;UAS-NPFR-RNAi* flies was normal (PPL1-γ2α′1, Kruskal-Wallis, n = 30, p>0.9999). (**D**) The performance of *MB296B;UAS-*
*Figure 11 continued on next page*

*Figure 11 continued*

*sNPFR-RNAi* flies was impaired (PPL1-γ2α′1, Kruskal-Wallis, n = 30–44, p=0.0145). (**E**) The performance of *MB058B;UAS-NPFR-RNAi* flies was normal (PPL1-α′2α2, n = 30, p>0.9999). (**F**) The performance of *MB058B;UAS-sNPFR-RNAi* flies was normal (PPL1-α′2α2, Kruskal-Wallis, n = 30, p=0.4948). (**G**) The performance of *MB087C;UAS-NPFR-RNAi* flies was impaired (PAM-β′2a, Kruskal-Wallis, n = 30, p=0.0155). (**H**) The performance of *MB087C;UAS-sNPFR-RNAi* flies was impaired (PAM-β′2a, Kruskal-Wallis, n = 30, p=0.0162). (**I**) The performance of *MB630B;UAS-NPFR-RNAi* flies was impaired (PPL1-α3, Kruskal-Wallis, n = 30, p<0.0001). (**J**) The performance of *MB630B;UAS-sNPFR-RNAi* flies was impaired (PPL1-α3, Kruskal-Wallis, n = 30, p=0.0038). (**K**) The performance of *MB301B;UAS-NPFR-RNAi* flies was impaired (PAM-β2β′2a, Kruskal-Wallis, n = 25–30, p=0.0087). (**L**) The performance of *MB301B; UAS-sNPFR-RNAi* flies was impaired (PAM-β2β′2a, Kruskal-Wallis, n = 30, p=0.0013). (**M**) The performance of *MB320C;UAS-dInR-RNAi* flies was normal (PPL1-γ1pedc, Kruskal-Wallis, n = 30, p>0.9999). (**N**) The performance of *MB320C0;UAS-DAR1-RNAi* flies was normal (PPL1-γ1pedc, Kruskal-Wallis, n = 30, p>0.9999). (**O**) The performance of *MB296B;UAS-dInR-RNAi* flies was enhanced (PPL1-γ2α′1, Kruskal-Wallis, n = 24–25, p=0.0199). (**P**) The performance of *MB296B;UAS-DAR1-RNAi* flies was enhanced (PPL1-γ2α′1, Kruskal-Wallis, n = 30, p=0.0002). (**Q**) The performance of *MB058B;UAS-dInR-RNAi* flies was enhanced (PPL1-α′2α2, Kruskal-Wallis, n = 29–30, p=0.0126). (**R**) The performance of *MB058B;UAS-DAR1-RNAi* flies was enhanced (PPL1-α′2α2, Kruskal-Wallis, n = 30, p=0.009). (**S**) The performance of *MB087C;UAS-dInR-RNAi* flies was enhanced (PAM-β′2a, Kruskal-Wallis, n = 30, p=0.0192). (**T**) The performance of *MB087C;UAS-DAR1-RNAi* flies was normal (PAM-β′2a, Kruskal-Wallis, n = 30, p=0.3645). (**U**) The performance of *MB630B;UAS-dInR-RNAi* flies was normal (PPL1-α3, Kruskal-Wallis, n = 30, p=0.6620). (**V**) The performance of *MB630B;UAS-DAR1-RNAi* flies was normal (PPL1-α3, Kruskal-Wallis, n = 30, p=0.6699). (**W**) The performance of *MB301B;UAS-dInR-RNAi* flies was normal (PAM-β2β′2a, Kruskal-Wallis, n = 25–30, p>0.9999). (**X**) The performance of *MB301B;UAS-DAR1-RNAi* flies was enhanced (PAM-β2β′2a, Kruskal-Wallis, n = 30, p=0.0071). Individual data points and mean ± SEM are shown.

DOI: https://doi.org/10.7554/eLife.35264.031

The following figure supplement is available for figure 11:

**Figure supplement 1.** A second set of RNAi lines confirms that the six assessed DANs are regulated by combinations of hunger and satiety signals.
DOI: https://doi.org/10.7554/eLife.35264.032

## Odor specificity in MB-mediated behavior

The lack of stereotypic connectivity between olfactory inputs and the KCs, together with the reinforcing DANs and the valence-encoding MBONs, make the MB an ideal center for olfactory learning (*Owald and Waddell, 2015*). It has been suggested that a given odor drives a collection of balanced MBON outputs in naive flies, and learning activates distinct reinforcing DANs that skew the outputs by changing the connectivity of specific KC-to-MBON synapses (*Aso et al., 2014b*; *Owald and Waddell, 2015*). Although each MBON and DAN is in contact with a large number of KCs wherein odors are sparsely coded, learning only modifies the output synapses of the KCs that are activated near to when reinforcement signals are triggered. Therefore, learning only skews the MBON outputs driven by trained odors. Such odor specificity is more difficult to achieve in MB-mediated motivated behavior. Hunger seems to regulate DAN activity and tip the balance of MBON outputs independently of olfactory inputs. According to the current model of how the MB circuit operates, starvation should assign positive valence to all odors, rather than just food odors. However, this supposition may not be entirely true. It has been shown that the MB is dispensable for $CO_2$ avoidance in food-satiated flies but, when flies are starved, blockage of the KCs decreases $CO_2$ avoidance (*Bräcker et al., 2013*). Therefore, starvation in this scenario seems to change the $CO_2$-driven MBON output valence from nil to negative. Interestingly, a comprehensive study on odor tuning in MBONs suggests that although MBONs are generally broadly tuned to odors, they do encode some odor specificity. In particular, odor groups of opposing valence are well separated in the MBON coding space (*Hige et al., 2015b*). How this is achieved given the probabilistic input connectivity of the antennal lobe projection neurons to the KCs (*Caron et al., 2013*) remains to be answered.

## Materials and methods

### Key resources table

| Reagent type | Designation | Source or reference | Identifiers | Additional information |
|---|---|---|---|---|
| Fly line | *E0067-Gal4* | (*Pai et al., 2013*) | NA | Gift from Ann-Shyn Chiang |
| Fly line | *G0239-Gal4* | (*Pai et al., 2013*) | Flybase: FBti0132502 | |
| Fly line | *MB002B-SplitGal4* | (*Aso et al., 2014a*) | RRID:BDSC_68305 | |
| Fly line | *MB005B-SplitGal4* | (*Aso et al., 2014a*) | RRID:BDSC_68306 | |

*Continued on next page*

*Continued*

| Reagent type | Designation | Source or reference | Identifiers | Additional information |
|---|---|---|---|---|
| Fly line | MB008B-SplitGal4 | (*Aso et al., 2014a*) | RRID:BDSC_68291 | |
| Fly line | MB011B-SplitGal4 | (*Aso et al., 2014a*) | RRID:BDSC_68294 | |
| Fly line | MB018B-SplitGal4 | (*Aso et al., 2014a*) | RRID:BDSC_68296 | |
| Fly line | MB027B-SplitGal4 | (*Aso et al., 2014a*) | RRID:BDSC_68301 | |
| Fly line | MB050B-SplitGal4 | (*Aso et al., 2014a*) | RRID:BDSC_68365 | |
| Fly line | MB051B-SplitGal4 | (*Aso et al., 2014a*) | RRID:BDSC_68275 | |
| Fly line | MB052B-SplitGal4 | (*Aso et al., 2014a*) | NA | Gift from Yoshinori Aso |
| Fly line | MB057B-SplitGal4 | (*Aso et al., 2014a*) | RRID:BDSC_68277 | |
| Fly line | MB062C-SplitGal4 | (*Aso et al., 2014a*) | NA | Gift from Yoshinori Aso |
| Fly line | MB077B-SplitGal4 | (*Aso et al., 2014a*) | RRID:BDSC_68283 | |
| Fly line | MB077C-SplitGal4 | (*Aso et al., 2014a*) | RRID:BDSC_68284 | |
| Fly line | MB080C-SplitGal4 | (*Aso et al., 2014a*) | RRID:BDSC_68285 | |
| Fly line | MB082C-SplitGal4 | (*Aso et al., 2014a*) | RRID:BDSC_68286 | |
| Fly line | MB083C-SplitGal4 | (*Aso et al., 2014a*) | RRID:BDSC_68287 | |
| Fly line | MB085C-SplitGal4 | (*Aso et al., 2014a*) | RRID:BDSC_68288 | |
| Fly line | MB090C-SplitGal4 | (*Aso et al., 2014a*) | NA | Gift from Yoshinori Aso |
| Fly line | MB091C-SplitGal4 | (*Aso et al., 2014a*) | NA | Gift from Yoshinori Aso |
| Fly line | MB093C-SplitGal4 | (*Aso et al., 2014a*) | RRID:BDSC_68289 | |
| Fly line | MB110C-SplitGal4 | (*Aso et al., 2014a*) | RRID:BDSC_68262 | |
| Fly line | MB112C-SplitGal4 | (*Aso et al., 2014a*) | RRID:BDSC_68263 | |
| Fly line | MB131B-SplitGal4 | (*Aso et al., 2014a*) | RRID:BDSC_68265 | |
| Fly line | MB210B-SplitGal4 | (*Aso et al., 2014a*) | RRID:BDSC_68272 | |
| Fly line | MB242A-SplitGal4 | (*Aso et al., 2014a*) | RRID:BDSC_68307 | |
| Fly line | MB262B-SplitGal4 | (*Aso et al., 2014a*) | RRID:BDSC_68254 | |
| Fly line | MB298B-SplitGal4 | (*Aso et al., 2014a*) | RRID:BDSC_68309 | |
| Fly line | MB310C-SplitGal4 | (*Aso et al., 2014a*) | RRID:BDSC_68313 | |
| Fly line | MB399B-SplitGal4 | (*Aso et al., 2014a*) | RRID:BDSC_68369 | |
| Fly line | MB433B-SplitGal4 | (*Aso et al., 2014a*) | RRID:BDSC_68324 | |
| Fly line | MB434B-SplitGal4 | (*Aso et al., 2014a*) | RRID:BDSC_68325 | |
| Fly line | MB542B-SplitGal4 | (*Aso et al., 2014a*) | RRID:BDSC_68372 | |
| Fly line | MB543B-SplitGal4 | (*Aso et al., 2014a*) | RRID:BDSC_68335 | |
| Fly line | MB549C-SplitGal4 | (*Aso et al., 2014a*) | RRID:BDSC_68373 | |
| Fly line | MB622B-SplitGal4 | (*Aso et al., 2014a*) | NA | Gift from Yoshinori Aso |
| Fly line | R21D02-Gal4 | (*Owald et al., 2015*) | RRID:BDSC_48939 | |
| Fly line | R50H05-Gal4 | (*Albin et al., 2015*) | RRID:BDSC_38764 | |
| Fly line | R66C08-Gal4 | (*Owald et al., 2015*) | RRID:BDSC_49412 | |
| Fly line | VT1211-Gal4 | (*Owald et al., 2015*) | VDRC: 202324 | |
| Fly line | VT999036-Gal4 | (*Aso et al., 2014a*) | NA | Gift from Yoshinori Aso |
| Fly line | MB320C-SplitGAL4 | (*Aso et al., 2014a*) | RRID:BDSC_68253 | |
| Fly line | MB058B-SplitGal4 | (*Aso et al., 2014a*) | RRID:BDSC_68278 | |
| Fly line | MB087C-SplitGal4 | (*Aso et al., 2014a*) | RRID:BDSC_68366 | |
| Fly line | MB296B-SplitGal4 | (*Aso et al., 2014a*) | RRID:BDSC_68308 | |
| Fly line | MB301B-SplitGal4 | (*Aso et al., 2014a*) | RRID:BDSC_68311 | |
| Fly line | MB630B-SplitGal4 | (*Aso et al., 2014a*) | RRID:BDSC_68334 | |

*Continued on next page*

Continued

| Reagent type | Designation | Source or reference | Identifiers | Additional information |
|---|---|---|---|---|
| Fly line | *UAS-DAMBi | (Pimentel et al., 2016) | VDRC: 105324; RRID:FlyBase_FBst0478846 | |
| Fly line | *UAS-GADi | (Barnstedt et al., 2016) | VDRC: 32344; RRID:FlyBase_FBst0459538 | |
| Fly line | *UAS-Rdli | (Cheung and Scott, 2017) | VDRC: 41103; RRID:FlyBase_FBst0463935 | |
| Fly line | UAS-5HT1Ai | (Lee et al., 2011) | VDRC: 106094; RRID:FlyBase_FBst0472248 | |
| Fly line | *UAS-5HT1Bi | (Mohammad et al., 2016) | VDRC: 109929; RRID:FlyBase_FBst0476393 | |
| Fly line | *UAS-5HT2Ai | (Mohammad et al., 2016) | VDRC: 102105; RRID:FlyBase_FBst0475582 | |
| Fly line | UAS-5HT2Bi | (Mohammad et al., 2016) | VDRC: 102356; RRID:FlyBase_FBst0478281 | |
| Fly line | UAS-5HT7i | (Dietzl et al., 2007) | VDRC: 104804; RRID:FlyBase_FBst0472736 | |
| Fly line | *UAS-NPFRi | (Ni et al., 2009) | RRID:BDSC_25939 | |
| Fly line | *UAS-sNPFRi | (Hong et al., 2012) | VDRC: 38925; RRID:FlyBase_FBst0462758 | |
| Fly line | *UAS-dInRi | (Loh et al., 2017) | RRID:BDSC_35251 | |
| Fly line | *UAS-DAR1i | (Yamagata et al., 2016) | NA | Gift from Hiromu Tanimoto |
| Fly line | *UAS-DAMBi$^2$ | (Hattori et al., 2017) | RRID:BDSC_51423 | |
| Fly line | *UAS-GADi$^2$ | (Koganezawa et al., 2016) | RRID:BDSC_51794 | |
| Fly line | *UAS-Rdli$^2$ | (Koganezawa et al., 2016) | RRID:BDSC_52903 | |
| Fly line | *UAS-5HT1Bi$^2$ | (Kaneko et al., 2017) | RRID:BDSC_33418 | |
| Fly line | *UAS-5HT2A$^2$ | (Ni et al., 2009) | RRID:BDSC_56870 | |
| Fly line | *UAS-NPFRi$^2$ | (Dietzl et al., 2007) | VDRC: 107663; RRID:FlyBase_FBst0481454 | |
| Fly line | *UAS-dInRi$^2$ | (Ni et al., 2009) | RRID:BDSC_51518 | |
| Fly line | *UAS-sNPFRi$^2$ | (Hu et al., 2017) | RRID:BDSC_27507 | |
| Fly line | *UAS-DAR1i$^2$ | (Yamagata et al., 2016) | NA | Gift from Hiromu Tanimoto |
| Fly line | Orco$^2$ | (Larsson et al., 2004) | RRID:BDSC_23130 | |
| Fly line | IR25a$^2$ | (Abuin et al., 2011) | RRID:BDSC_41737 | |
| Fly line | IR8a$^1$ | (Abuin et al., 2011) | RRID:BDSC_41744 | |
| Fly line | DAMB | (Selcho et al., 2009) | Flybase: FBab0047678 | |
| Fly line | DopR1 (dumb$^2$) | (Qin et al., 2012) | Exelixis: f02676; RRID:FlyBase_FBst1017920 | |
| Fly line | D2R$^{f06521}$ | (Marella et al., 2012) | Exelixis: f06521; RRID:FlyBase_FBst1020637 | |
| Fly line | DopEcR$^{c02142}$ | (Inagaki et al., 2012) | Exelixis: c02142; RRID:FlyBase_FBst1006135 | |
| Fly line | UAS-TrpA1 | (Hamada et al., 2008) | Flybase: FBtp0040248 | Gift from Scott Waddell |
| Fly line | UAS-shi$^{ts1}$ | (Kitamoto, 2001) | Flybase: FBtp0013545 | Gift from Scott Waddell |
| Fly line | UAS-mCD8-GFP | (Lee and Luo, 1999) | Flybase: FBtp0002652 | Gift from Scott Waddell |
| Fly line | UAS-GCaMP6m | (Chen et al., 2013) | RRID:BDSC_42748 | |
| Fly line | UAS-DenMark,UAS-Dsyd-1::GFP | (Owald et al., 2015) | NA | Gift from Scott Waddell |
| Fly line | nSyb-GAL4 | (Pauli et al., 2008) | Flybase: FBtp0041245 | Gift from Scott Waddell |
| Fly line | da-GAL4 | (Wang et al., 2017) | RRID:BDSC_55851 | |

*Continued*

| Reagent type | Designation | Source or reference | Identifiers | Additional information |
|---|---|---|---|---|
| Antibody | Mouse anti-brp (nc82) | Developmental Studys Hybridoma Bank (DSHB), IA, USA | RRID:AB_2314866 | |
| Antibody | Chicken anti-GFP | Abcam, UK | RRID:AB_300798 | 1:5000 |
| Antibody | Rabbit anti-Dsred | Clontech, CA, USA | RRID:AB_10013483 | 1:500 |
| Antibody | Rat anti-mCD8α | Thermo Fisher Scientific, MA, USA | RRID:AB_10392843 | 1:100 |
| Antibody | Goat anti-rabbit (Cy3) | Jackson ImmunoResearch, PA, USA | RRID:AB_2338006 | 1:400 |
| Antibody | Goat anti-mouse (Cy3) | Jackson ImmunoResearch, PA, USA | RRID:AB_2338692 | 1:400 |
| Antibody | Donkey anti-chicken (Alexa 488) | Jackson ImmunoResearch, PA, USA | RRID:AB_2340375 | 1:400 |
| Antibody | Goat anti-rat (Alexa 488) | Thermo Fisher Scientific, MA, USA | RRID:AB_141373 | 1:400 |
| Food odor | Dry yeast | Ferminpan red, Italy | NA | |
| Food odor | Apple cider vinegar | Alce Nero, Italy | NA | |
| Food odor | Banana powder | Gen Asia Biotech, Taiwan | NA | |
| Chemical | Sucrose | Merck, Germany | Cat# 107687 | |
| Chemical | Agar | BD, NJ, USA | Cat# 214530 | |
| Chemical | Formaldehyde | Sigma, MO, USA | Cat# F8775 | |
| Chemical | PBS | Sigma, MO, USA | Cat# P4417 | |
| Chemical | Gold Antifade reagent | Thermo Fisher Scientific, MA, USA | Cat# S36937 | |
| Chemical | paraffin wax | Sigma, MO, USA | Cat# 327304 | |
| Serum | Normal goat serum | Jackson Immuno Research, PA, USA | RRID:AB_2336990 | |
| Kits | TRIzol RNA Isolation reagents | Thermo Fisher Scientific, MA, USA | Cat# 15596026 | |
| Kits | SuperScript IV First-Strand Synthesis System for RT-PCR Kit | Thermo Fisher Scientific, MA, USA | Cat# 18091050 | |
| Software | Prism 7 | GraphPad, CA, USA | RRID:SCR_002798 | |
| Software | MATLAB 2017a | MathWorks, MA, USA | RRID:SCR_001622 | |
| Software | Fiji/ImageJ | Fiji | RRID:SCR_002285 | |

*Asterisks indicate the RNAi lines whose knockdown efficiencies are shown in *Figure 6—figure supplement 3*

## Fly strains

Fly strains used in this study are listed in Key resources table. Flies were reared on standard corn-meal food and a 12 hr light:12 hr dark cycle. All flies were raised at 23°C and 60% humidity unless stated otherwise.

## Food-seeking assay

Single-fly assays were used to measure food-finding performances of flies (*Video 1*). Male flies (5–7 days old) of appropriate genotype were collected under $CO_2$ anesthesia and allowed to recover for 2 days on standard cornmeal food. Single flies were introduced into petri dishes (85 mm in diameter and 6 mm in height) close to the dish wall. For experiments conducted at 23°C, a 5 μl drop of yeast, apple cider vinegar (Alce Nero, Italy), or banana odor solution was placed in the middle of each dish. The banana odor solution was prepared by dissolving 3 g banana powder (Gen Asia Biotech, Taiwan) in 5 ml sterilized water. The yeast solution was prepared as follows: 1 g dry yeast (Ferminpan red, Italy) and 5 g sucrose (Merck, Germany; 107687) were mixed with 50 ml sterilized

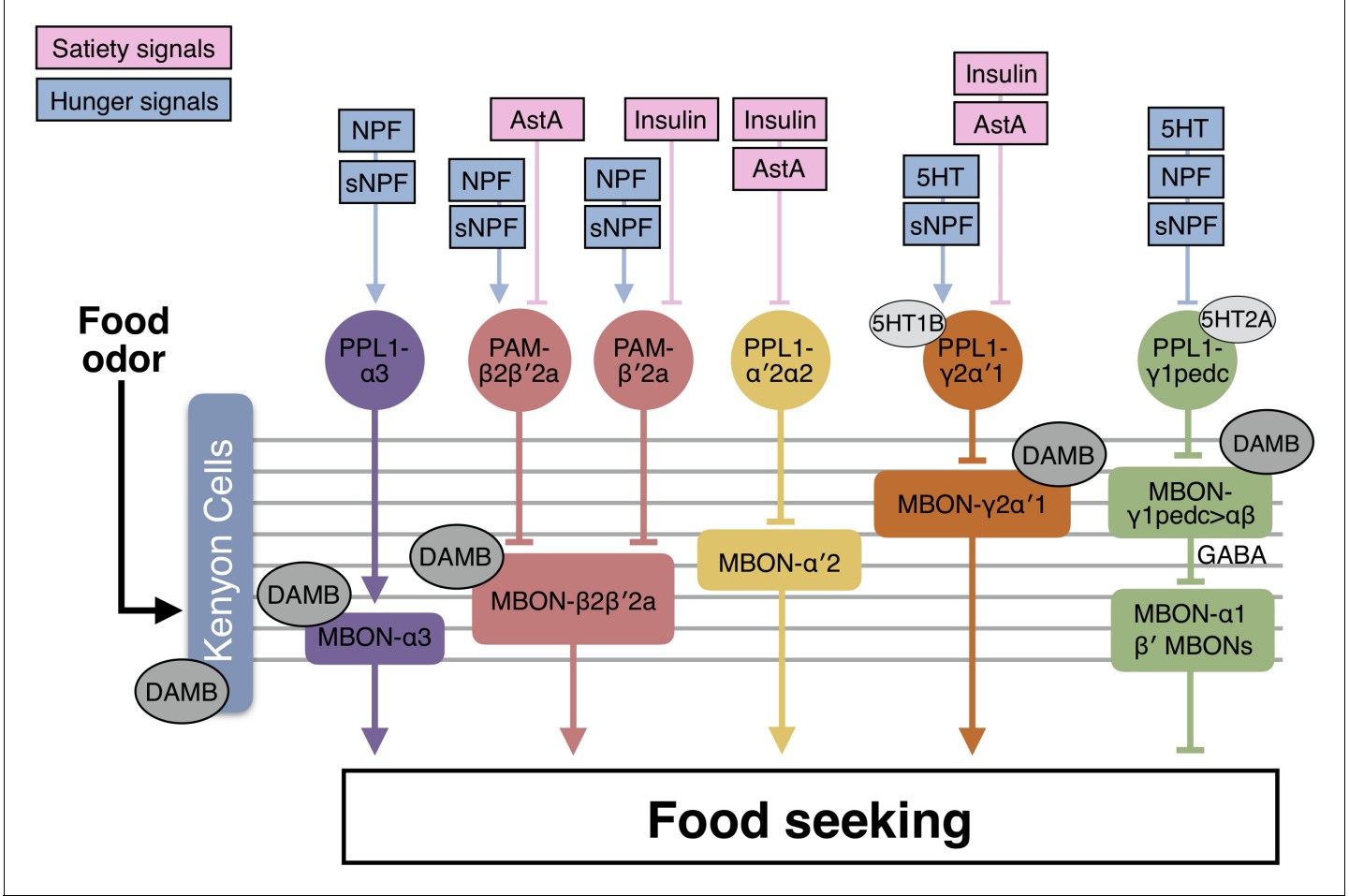

**Figure 12.** A model for the neural mechanics of the MB circuit in controlling food-seeking behavior. During food seeking, odors activate the KCs and in turn activate MBON-α3, MBON-β2β'2a, MBON-α'2, MBON-γ2α'1 and MBON-γ1pedc>αβ. GABAergic MBON-γ1pedc>αβ inhibits the downstream neurons that suppress food-seeking behavior, including β'2-innervating MBONs and MBON-α1. KC-to-MBON connectivity is regulated by the corresponding DANs. The DANs are regulated by combinations of hunger and satiety signals. When flies are food-satiated, satiety signals like insulin and AstA suppress PPL1-γ2α'1, PPL1-α'2α2, PAM-β'2a, and PAM-β2β'2a DANs. When flies are starved, hunger signals including serotonin (5HT), NPF, and sNPF activate PPL1-α3, PAM-β2β'2a, PAM-β'2a, and PPL1-γ2α'1 DANs, whereas they suppress PPL1-γ1pedc DANs. Dopamine signals pre- and post-synaptically mediated by the DAMB receptor fine-tune the KC-to-MBON connectivity and modulate the collective output of the MBONs driven by food odor. Therefore, hunger state tunes the odor-driven output of the MBONs to regulate food-seeking behavior.

DOI: https://doi.org/10.7554/eLife.35264.033

The following figure supplement is available for figure 12:

**Figure supplement 1.** A hypothetical model for the MBONs that show reduced responses to yeast odor in hungry flies.
DOI: https://doi.org/10.7554/eLife.35264.034

water and incubated in a 28°C shaking incubator (170 rpm) for 16 hr. For experiments conducted at 32°C, the odor solutions were diluted (1:4 for yeast; 1:500 for apple cider vinegar; 1:3 for banana odor) by mixing with 1% agar (BD, NJ, USA; 214530) because odors are more volatile and evaporate more easily at higher temperature. At these dilutions, wild-type flies exhibited similar food-seeking performance under both temperatures. Experiments were conducted under 630 nm LED lights. A fly was considered as having found the food when it rested for 3 s or longer on the food drop. We chose 3 s to avoid scoring flies that accidentally passing by the food drop. To induce hunger in flies, they were starved on 1% agar for 24 hr, except for the experiments described in *Figure 1A*, in which they were starved for various durations. For experiments using *UAS-TrpA1* and *UAS-Shi^{ts1}*, flies were raised at 23°C before being shifted to 32°C for 10 min to activate or block the relevant neurons. All experiments not involving a temperature-sensitive effector were performed at 23°C. We calculated

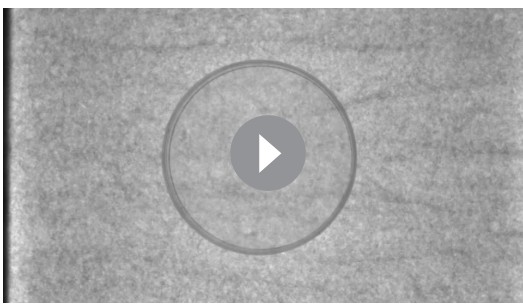

**Video 1.** Single fly food-seeking assay
DOI: https://doi.org/10.7554/eLife.35264.035

the 'Food-seeking index' as: [Total assay time (600 s) - the time taken to locate food (sec)]/Total assay time (600 s). Statistical analyses were conducted in Prism 7 software (GraphPad, CA, USA; RRID:SCR_002798). As our behavioral data did not conform to normal distributions, Kruskal-Wallis and Dunn's multiple comparison tests were used to detect differences between experimental groups and their relevant controls.

## Conditioning flies before food-seeking assay

Around 50 flies with desired genotypes were starved for 24 hr on 1% agar at 23°C. The flies were then transferred into a 32°C room for 8 min before being loaded into the training tube of a T-maze (CelExplorer Labs Co., Taiwan; TM-101). The flies were presented with yeast odor (prepared as for

**Table 2.** Primers used for RT-PCR

| Primer list | Primer sequence (5′ → 3′) |
| --- | --- |
| sNPFR-F | CCAACTGGAGCCTAACGTCG |
| sNPFR-R | AACTGGTTGTGAATGATCCCG |
| 5HT1B-F | TTGGTTGCATCTCTGGCAGTG |
| 5HT1B-R | CCGGTCCCAATATCCATCCATT |
| 5HT2A-F | TTCACACTGCGACACTTCAAT |
| 5HT2A-R | GGGGTGTAGGATGTGCTGT |
| InR-F | CCGCAAGCAGTGAAGAAGC |
| InR-R | CGTCGTCTCCACTTCGTCAAA |
| DAR1-F | CCCGTATTCTTTGGCATTATCGG |
| DAR1-R | GGCCAGGTTGATTATCAGCAGA |
| RDL-F | CACAGGCAACTATTCGCGTTT |
| RDL-R | GCGATTGAGCCAAAATGATACC |
| GAD-F | CACCAACGACCGGAACGAG |
| GAD-R | TGGGGATGTCCCGTCTTAACT |
| DAMB-F | CATCTCCGAGGATGTCTACTTCT |
| DAMB-R | CCATCGCAGGACTCAAGGTG |
| NPFR-F | ATCAGCATGAATCAGACGGAGC |
| NPFR-R | GATGCCGGTCGTCCAGATA |
| Rpl19-F | TCTCTAAAGCTCCAGAAGAGGC |
| Rpl19-R | CGATCTCGTTGATTTCATTGGGA |

DOI: https://doi.org/10.7554/eLife.35264.036

the single-fly behavior assay) for 2 min and then immediately transferred to vials at 23°C for 3 min and tested for food-seeking behavior.

## Measuring fly moving speed

Single flies were transferred into petri dishes (85 mm in diameter and 6 mm in height) backlit by a red LED panel (630 nm) and allowed to become accustomed to the environment for 3 min. The movement of the flies was then recorded for 2.5 min at 20 fps using a camera (Basler, Germany; acA2040-90 um) from above. The videos were analyzed using custom-made MATLAB programs (Mathworks, MA, USA) (*Source code 1* and *2*).

## Immunofluorescence staining

Fly brains were dissected in 1X PBS (Sigma; P4417), and fixed in PBS containing 4% formaldehyde (Sigma, MO, USA; F8775) for 20 min at room temperature. After fixation, the brains were washed three times for 20 min each in PBST (0.5% Triton X-100 in PBS) and incubated for 30 min in PBST with 5% normal goat serum (blocking solution; Jackson ImmunoResearch, PA, USA; RRID:AB_2336990). Then the brains were incubated in the blocking solution with primary antibodies at 4°C overnight. The next day, the brains were washed three times for 20 min each in PBST at room temperature and then incubated with secondary antibodies in PBST at 4°C overnight. The brains were then washed three times for 20 min each in PBST at room temperature and mounted with Gold Antifade reagent (Thermo Fisher Scientific, MA, USA; S36937). Antibodies used in this study are listed in Key resources table. The stained brains were imaged using a confocal microscope (Zeiss LSM880) and analyzed in the software platform Fiji/ImageJ (RRID:SCR_002285). To measure the DSyd-1-GFP and DenMark signals, regions of interest (ROIs) containing DSyd-1-GFP were manually outlined focal plane by focal plane. Mean intensities of both DSyd-1-GFP and DenMark signals were measured using the same ROIs. These mean intensity data are normally distributed (D'Agostino and Pearson normality test, p>0.05) and therefore comparisons between fed and hungry flies were made using unpaired t-test.

## In vivo functional imaging

Two-photon imaging of odor-evoked calcium responses was performed on 3–8 day-old flies following 22–26 hr of starvation or *ad libitum* feeding. Flies were anesthetized on ice and mounted in an imaging chamber (Warner Instruments, CT, USA; PH-5/RC-20), and the head was affixed by sealing the eyes to the chamber with paraffin wax (Sigma; 327304). The legs and proboscis were immobilized with wax to reduce movements while imaging. Part of the head capsule was removed to allow optic access to the brain under sugar-free HL3-like saline (*Yoshihara, 2012*). We used a 20x water-immersion objective and a Zeiss 880 upright laser scanning confocal microscope with a two-photon laser (Spectra-Physics, CA, USA; Mai Tai HP 1040S) to acquire images (100 × 100 pixels; 8.088 Hz). Yeast odor was delivered on a clean air carrier stream, and it was diluted further x10 or x100 with sterilized ddH$_2$O for some experimental groups. At the resting state, air constantly flowed through the control vial and was delivered to the fly antennae. During imaging, flies were exposed to yeast odor for 10 s, then re-exposed to clean air for the remaining time. At least two trials were conducted on each fly, and the inter-trial interval was longer than 3 mins. For data analysis, all acquired images were analyzed using Fiji/ImageJ (RRID:SCR_002285). Regions of interest were manually assigned to the anatomically distinct neuronal processes. The change in ΔF/F was calculated, with baseline fluorescence F being defined as the mean fluorescence from 5 s prior to odor delivery. Data from individual flies is presented as the mean result from at least two trials. The area under the curve (AUC) was measured as the integral of ΔF/F during the 10 s between onset and offset of odor delivery. Data were excluded if flies did not show any visible response.

## RT-PCR

RT-PCR was performed using a S1000 Thermal Cycler (BioRad, CA, USA). Total RNA from dissected adult brains (for *DAMB*, *Rdl*, *sNPFR*, *NPFR*, *DAR1*, *5HT1B*, *5HT2A*) or third-instar larvae (for *GAD* and *InR*) was isolated using TRIzol RNA Isolation reagents (Thermo Fisher Scientific; 15596026), followed by reverse transcription of cDNAs using SuperScript IV First-Strand Synthesis System for RT-PCR Kit (Thermo Fisher Scientific; 18091050). Sequences specific for the genes of interests were PCR

amplified using specific primer pairs (*Table 2*). Twenty amplification cycles were conducted for *Rdl* and *dInR* and their controls, and 30 for the rest. The intensity of PCR bands was quantified using Fiji/ImageJ and normalized to the internal control of *ribosomal protein L19* (*Rpl19*) mRNA.

## Acknowledgements

We thank Wei-Fang Liu for assisting with some of the behavioral experiments, and the Cheng-Ting Chien Lab (Academia Sinica, Taiwan) for sharing fly lines and providing help with RT-PCR. We thank Scott Waddell (University of Oxford, UK), Yoshinori Aso (Janelia Farm Research Campus, USA), Ann-Shyn Chiang (National Tsing Hua University, Taiwan), Hiromu Tanimoto (Tohoku University, Japan), Andreas Thum (University of Konstanz, Germany), Joshua Dubnau (Stony Brook University, USA), and Chia-Lin Wu (Chang Gung University, Taiwan) for sharing fly lines. We also thank Andrew Lin (University of Sheffield, UK), Emmanuel Perisse (University of Oxford, UK), and Gaurav Das (National Centre for Cell Science, India) for commenting on the manuscript. Thanks are also due to the Bloomington Stock Center, the Vienna Drosophila RNAi Center, the Exelixis collection at Harvard Medical School, and the Fly Core in Taiwan for providing fly stocks.

## Additional information

### Funding

| Funder | Grant reference number | Author |
| --- | --- | --- |
| Ministry of Science and Technology, Taiwan | 105-2628-B-001-005-MY3 | Suewei Lin |
| Academia Sinica | | Suewei Lin |

The funders had no role in study design, data collection and interpretation, or the decision to submit the work for publication.

### Author contributions

Chang-Hui Tsao, Chien-Chun Chen, Conceptualization, Data curation, Formal analysis, Validation, Investigation, Visualization, Methodology, Writing—review and editing; Chen-Han Lin, Data curation, Validation, Investigation, Visualization, Writing—review and editing; Hao-Yu Yang, Software, Validation, Investigation, Methodology, Writing—review and editing; Suewei Lin, Conceptualization, Data curation, Supervision, Investigation, Visualization, Methodology, Writing—original draft, Project administration, Writing—review and editing

### Author ORCIDs

Suewei Lin [iD] http://orcid.org/0000-0001-7079-7818

### Decision letter and Author response

Decision letter https://doi.org/10.7554/eLife.35264.041
Author response https://doi.org/10.7554/eLife.35264.042

## Additional files

### Supplementary files

• Source code 1. MATLAB program used to generate the moving trajectory of a recorded fly.
DOI: https://doi.org/10.7554/eLife.35264.037

• Source code 2. MATLAB program used to analyze the moving speed of a recorded fly based on the trajectory generated by *Source code 1.*
DOI: https://doi.org/10.7554/eLife.35264.038

• Transparent reporting form
DOI: https://doi.org/10.7554/eLife.35264.039

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
