## [Decision Letter]

[Editors’ note: a previous version of this study was rejected after peer review, but the authors submitted for reconsideration. The first decision letter after peer review is shown below.]

Thank you for submitting your work entitled *Drosophila* mushroom bodies integrate hunger and satiety signals to control innate food-seeking behavior" for consideration by *eLife*. Your article has been reviewed by two peer reviewers, and the evaluation has been overseen by a Reviewing Editor and a Senior Editor. The reviewers have opted to remain anonymous.

The two reviewers agreed that examining the effects of hunger and satiety on the larger MB circuit is interesting and important. However, it is not clear how generalizable some of the results are and a number of additional experiments were recommended. Because these experiments are likely to require more than two months to complete, we feel that it is most appropriate to reject the manuscript. However, if the experiments suggested by reviewer 2 (1-3) are performed, we would be happy to consider a new version. Of course, we understand that you may wish to take your paper elsewhere at this time, and certainly would not wish you to carry out extensive new experiments unless you agree that they are scientifically reasonable.

*Reviewer #2:*

The mushroom body (MB) in insect brains has been extensively studied as a model system for learning and memory. Dr. Lin previously demonstrated that behavioral expression of odor-reward memories in the MB are gated by motivational states (i.e. hungry or thirsty). This paper provides comprehensive circuit-level analysis of the MB for innate behavioral responses to a food-related odor and its motivational regulation. The authors first established a simple assay to measure response time for flies to locate food, and showed that it requires olfaction and the response time is shortened when flies are hungry. Blocking any one of three major classes of MB's intrinsic neurons or specific MB output neurons (MBONs) prolonged response time. Using in vivo calcium imaging, the authors demonstrated that odor responses of MB output neurons to yeast are altered in hungry flies. Intriguingly the effect of starvation differed among the MBON cell types. Furthermore, the authors extended their analysis to dopaminergic neurons (DANs), connections between MBONs and molecular mechanisms.

Overall, the paper provides new insights into hunger modulation of the MB circuit, which has not been studied sufficiently. Below I list my major concerns and suggestions for further improvements.

1) The authors proposed that starvation and satiety signals modulate DANs, and that in turn modulates odor response of MBONs. Directly demonstrating starvation effects on DANs and Kenyon Cells is essential to support this idea. Also, odor concentration dependency in Figure 4B and E could be due to modulation at earlier steps (e.g. in the antennal lobes) rather than dopamine-dependent modulation in the MB lobes.

2) Show specificity and generality of the effect to food odors by testing other odors. Using T-maze, Bräcker et al., 2013 described that "Blocking of MB output, on the other hand, had no effect on the attraction to vinegar alone (Figure S1)". Repeating experiments in Figure 2 and 3 with vinegar (and if necessary repeating experiments in Figure 4 with vinegar) would be important to evaluate the generality of the findings.

3) dTrpA1 activation of the DANs could induce aversive memory of yeast. Recent papers showed that flies can form olfactory memory when they smelled odors while MBONs or DANs are blocked (Yamagata et al., 2016; Ueoka et al., 2017). Therefore, the observed defect in some of dTrpA1 shibire experiments (e.g. Figure 3A and 6F) could be because flies formed aversive memory of yeast odors. I suggest repeating at least a few key results in a learning mutant background; i.e., repeat shibire and dTrpA1 experiments in Figure 3 and 6 in *rutabaga2080* mutant background.

4) Description of the assay and quantification of flies' behavior are insufficient. Diagram of the assay, example trajectories of flies and movies would be helpful to better understand the assay.

5) Authors defined that a fly was considered as having found the food when it rested for 3 s or longer on the food drop. Why 3 seconds? What if it is set as 1s or shorter?

6) For all the behavioral experiments with RNAi, expression of receptors in the targeted cell types and its knockdown by RNAi should be shown. Otherwise, authors should significantly tone down conclusions regarding RNAi experiments, and explicitly write that RNAi experiment does not necessarily prove expression of the targeted gene.

*Reviewer #3:*

This manuscript extensively examined roles of the mushroom body (MB) －Kenyon cells (KCs), mushroom body output neurons (MBONs), and MB-projecting dopaminergic neurons (DANs) – in the yeast odor-seeking behavior in *Drosophila*.

Roles of the MB circuits in innate/acquired odor approach have repeatedly been reported (e.g. Bracker et al., 2013; Lewis et al., 2015; Owald et al., 2015, Nat Neurosci; Perisse et al., 2016). Additionally multiple studies have shown that hunger and satiety signals converge in the MB through DANs/damb, and change MBON responses (e.g. Krashes et al., 2009; Musso et al., 2015; Hige et al., 2015; Perisse et al., 2016). The major findings in this study is a) the comprehensive identification of cell types and the MB network important for yeast odor approach; b) that hunger/satiety changes the calcium responses of these MBONs to the yeast odor; c) neuropeptidergic pathways that modulate the dopaminergic neurons involved in yeast odor-seeking. Overall, the work is thorough and well controlled, and I have only few technical concerns.

1) I did not understand the authors' explanation regarding why the response of some MBONs required for the yeast-seeking are suppressed upon starvation (Figure 4).

2) This study used only yeast odor, and thus generalization to "food odors" should be done with more care; it is questionable if MBONs identified in this study are required for other food odors like vinegar.

3) The "food-seeking index" in their assay takes into account time spent for eating yeast. This means, in principle, that flies with normal odor approach but without continuous feeding (> 3s as the authors define) would be scored as "impaired" in food seeking. Were there such flies?

4) Maybe it might be too neurotic, but fly food usually contains yeast, such that the behavior in this assay can contain an experience-dependent component.

[Editors’ note: what now follows is the decision letter after the authors submitted for further consideration.]

Thank you for resubmitting your work entitled "*Drosophila* mushroom bodies integrate hunger and satiety signals to control innate food-seeking behavior" for further consideration at *eLife*. Your revised article has been favorably evaluated by Eve Marder (Senior Editor) and two reviewers, one of whom is a member of our Board of Reviewing Editors.

The manuscript has been improved but there are some remaining changes to the text that need to be addressed before acceptance, as outlined below:

1) Based on RNAi experiments, the authors concluded that "DANs receive surprisingly rich inputs of hunger and satiety signals". To support the conclusion, authors would need to show a) which specific serotonergic or peptidergic neurons mediate hunger/satiety signals, b) they are anatomically and functionally connected to DANs, c) localize receptors in DAN's dendrites, and d) establish physiological effect of serotonin/neuropeptides on DANs. Behavioral phenotype with RNAi may provide good starting point for such endeavor, but I would describe it as a modest evidence for "rich inputs of hunger and satiety signals". Please tone down the conclusions and discuss the caveats of these experiments.

2) Related to the above criticism, GABA receptor knock down in Figure 6B-D is consistent with the model in Figure 6F, but does not necessarily prove it. It is unclear if the RNAi knock down phenotype is due to disruption of GABA signals from MB-MVP2 or other GABAergic neurons including MB-APL. Please note this caveat.

3) The new Syd1-GFP data is possibly interesting, but it is very weak evidence for state dependent modulation of active zones. Syd1-GFP is under the control of GAL4/UAS rather than endogenous promoter as in brp-GFP BAC (Chen et al., 2014). Therefore GFP signals do not necessarily correlate with amount of endogenous Syd1. Also Syd1-GFP fluorescence would not change depending on whether Syd1-GFP molecules are in the functional active zone molecular complex or outside of active zone as a non-functional form. Authors would need more direct measurement of active zones quantity and density. Please state this caveat.

4) In Figure 5 and 9, it would be more intuitive if color for fed and starved are swapped. The color for "starved" should be more salient than basal "fed" state.

5) Figure 14 explains how the MBON that showed reduced response at hunger induces food-seeking behavior, but this can't be generalized to all the MBONs. In other words, it looks as if there were no other circuit explanations. This figure should be a figure supplement and the authors can also try to shorten the text explaining this model.

6) Table 2 shows extreme efforts that the authors have made, but is very hard to follow. Please make it simpler by showing the results by the cell types but not drivers.

7) The newly added Figure 4 has too much information. Reconsider the essential dataset to show in the main figure and move the others (e.g. the permissive control to a figure supplement).

---

## [Author Response]

[Editors’ note: the author responses to the first round of peer review follow.]

Reviewer #2:[…] Overall, the paper provides new insights into hunger modulation of the MB circuit, which has not been studied sufficiently. Below I list my major concerns and suggestions for further improvements.1) The authors proposed that starvation and satiety signals modulate DANs, and that in turn modulates odor response of MBONs. Directly demonstrating starvation effects on DANs and Kenyon Cells is essential to support this idea. Also, odor concentration dependency in Figure 4B and E could be due to modulation at earlier steps (e.g. in the antennal lobes) rather than dopamine-dependent modulation in the MB lobes.

We reason that the bi-directional changes in the MBON responses to yeast odor are most likely due to modulations of the KC-to-MBON synapses, given that a general increase or decrease in KC responses to odor should cause a general increase or decrease in MBON responses rather than compartment-specific changes. Beshel and Zhong have shown that, when measured from the cell bodies of all KCs using calcium imaging, the food odor-evoked response did not differ between fed and starved flies (Beshel and Zhong, 2013). In our revised manuscript, we have performed additional experiments and show that KC responses to yeast odor remained the same between fed and hungry flies even when we measured odorevoked calcium signals from specific MB lobe compartments innervated by the five food-seeking MBONs (revised Figure 5—figure supplement 1). These new data support that the hunger-modulated odor responses in the MBONs are not due to changes of odor response in KCs and their upstream circuits.

We agree with the reviewer that understanding which aspects of the DANs are modulated by starvation is an important and very interesting question. We have performed additional experiments to show that odor responses in some DANs are modulated by hunger. Interestingly, PPL1-γ1pdec DANs show a decreased odor response and PPL1-α3 DANs show an increased odor response in starved flies (revised Figure 9). This is consistent with our behavioral data, which suggest that hunger positively regulates PPL-α3 DANs but suppresses PPL1-γ1pdec DANs. Furthermore, in several DANs whose activities are promoted by starvation (as suggested by our behavioral data), the density or size of their active zones marked by Dsyd-1-GFP is significantly higher in hungry flies (revised Figure 10). Other physiological properties of the DANs may also be modulated by hunger and satiety, but a comprehensive survey is beyond the scope of the current study.

2) Show specificity and generality of the effect to food odors by testing other odors. Using T-maze, Bräcker et al., 2013 described that "Blocking of MB output, on the other hand, had no effect on the attraction to vinegar alone (Figure S1)". Repeating experiments in Figure 2 and 3 with vinegar (and if necessary repeating experiments in Figure 4 with vinegar) would be important to evaluate the generality of the findings.

We have added new data showing that both KCs and the five MBONs we have identified are also important for hungry flies to approach apple cider vinegar and banana odors (revised Figure 4). These data suggest that the MB pathways we have identified are generally required for flies to seek food odors. The difference between the T-maze results in Bräcker et al. and our data may highlight the importance of behavioral assay choice in uncovering neural mechanisms of behavior. In T-maze, flies are choosing between fresh air and high concentration of an odor, whereas in our assay, flies need to navigate along odor concentration gradients to locate food. We believe that our behavioral paradigm is a more natural setting for assaying food seeking behavior and is potentially more sensitive in identifying neurons required for this behavior.

3) dTrpA1 activation of the DANs could induce aversive memory of yeast. Recent papers showed that flies can form olfactory memory when they smelled odors while MBONs or DANs are blocked (Yamagata et al., 2016; Ueoka et al., 2017). Therefore, the observed defect in some of dTrpA1 shibire experiments (e.g. Figure 3A and 6F) could be because flies formed aversive memory of yeast odors. I suggest repeating at least a few key results in a learning mutant background; i.e., repeat shibire and dTrpA1 experiments in Figure 3 and 6 in rutabaga2080 mutant background.

The changes in food-seeking behavior we observed when we activated or blocked DANs are unlikely a result of olfactory learning. In the six DANs we have shown to regulate food-seeking behavior, four of them—PPL1-α3, PPL1-α′2α2, PPL1-γ2α′1, and PPL1-γ1pdec—have been shown to implant aversive olfactory memories when their activation is paired with odors (Aso and Rubin, 2016). In our study, we found that activation of PPL1-α3, PPL1-α′2α2, and PPL1-γ2α′1 DANs promotes rather than inhibits food-seeking behavior. Therefore, our findings are the opposite of what would be expected if the flies had formed aversive memories to yeast odors. The memory formation argument is plausible for PPL1-γ1pdec, and it is unclear what types of memory can be induced when pairing odors with the activation of PAM-β2β′2a and PAM-β′2a (the other two DANs that promote food-seeking behavior identified in our study). We have performed additional experiments to directly test whether artificially induced olfactory learning has any impact on our food-seeking assays. We paired yeast odor with the activation or silencing of PPL1-γ1pdec, PAM-β2β′2a, and PAM-β′2a DANs for 2 min and tested food-seeking behavior shortly thereafter. We found that this pre-conditioning of yeast odor has no effect on food-seeking behavior, arguing that olfactory learning plays a minimum role in our study (revised Figure 7—figure supplement 2). We have also done the same experiments for MBON-γ1pedc>αβ and found that pre-conditioning yeast odor with the silencing of MBON-γ1pedc>αβ has no effect on food-seeking behavior either (revised Figure 7—figure supplement 2). We did not perform the *rutabaga* experiment because we have shown that DAMB is critical for hunger-driven food-seeking behavior and the cAMP pathway has been shown to be downstream of DAMB (Himmelreich et al., Cell Report 2017; Han et al., Neuron 1996).

4) Description of the assay and quantification of flies' behavior are insufficient. Diagram of the assay, example trajectories of flies and movies would be helpful to better understand the assay.

We have added more detail to the description of our behavioral assay. We have also provided a movie of the assay.

5) Authors defined that a fly was considered as having found the food when it rested for 3 s or longer on the food drop. Why 3 seconds? What if it is set as 1s or shorter?

We used 3 s to reduce bias introduced by flies accidentally passing by the food drop and have added this point in the revised manuscript (subsections “Food-seeking assay” and “Flies approach yeast food when they are hungry”). We have performed additional experiments by scoring flies as on target whenever they touched the food drop and found that it does not affect our conclusions (Figure 3—figure supplement 3).

6) For all the behavioral experiments with RNAi, expression of receptors in the targeted cell types and its knockdown by RNAi should be shown. Otherwise, authors should significantly tone down conclusions regarding RNAi experiments, and explicitly write that RNAi experiment does not necessarily prove expression of the targeted gene.

We did not check the expression of receptors due to a lack of appropriate antibodies. However, most of the RNAi lines we used in our study have been published in other studies (see Table 1). Nevertheless, we have repeated all our RNAi knockdown experiments with a second set of RNAi lines and found the same results (revised Figure 6—figure supplement 1, Figure 8—figure supplement 1, Figure 11—figure supplement 3 and Figure 12—figure supplement 1). We have also included in this revised manuscript RT-PCR data showing the knockdown efficiency of the RNAi lines (revised Figure 6—figure supplement 3). I believe the replication of the behavioral phenotypes with this second set of RNAi lines greatly strengthens our conclusion that the DANs receive rich inputs of hunger and satiety signals. However, as suggested by the reviewer, we have added a note stating that expression of these receptors in the DANs remains to be established (subsection “DANs are differentially regulated by hunger and satiety signals”, last paragraph).

Reviewer #3:[…] 1) I did not understand the authors' explanation regarding why the response of some MBONs required for the yeast-seeking are suppressed upon starvation (Figure 4).

We have offered more explanations (subsection “Hunger tunes the response of MBONs to yeast odor”) and added a putative model (revised Figure 14) to illustrate how this might work.

2) This study used only yeast odor, and thus generalization to "food odors" should be done with more care; it is questionable if MBONs identified in this study are required for other food odors like vinegar.

We have repeated our MBON and KC experiments with vinegar and banana odors and found that these neurons are also required for flies to seek these two food odors (revised Figure 4). Therefore, the neural circuits we have identified here are likely generally involved in hunger and satiety control of food-odor seeking behavior.

3) The "food-seeking index" in their assay takes into account time spent for eating yeast. This means, in principle, that flies with normal odor approach but without continuous feeding (> 3s as the authors define) would be scored as "impaired" in food seeking. Were there such flies?

We did not observe such flies. Furthermore, we have performed additional experiments by scoring flies as on target whenever they touched the food drop and found that our conclusions remain the same (revised Figure 3—figure supplement 3).

4) Maybe it might be too neurotic, but fly food usually contains yeast, such that the behavior in this assay can contain an experience-dependent component.

We think that an experience-dependent component has a minimum effect on our behavioral assay for the following reasons. First, this assay also works for vinegar and banana odors, which are not part of our fly food recipe (revised Figure 4). Second, dDA1 has been shown to be essential for olfactory learning and memory, but dDA1 mutant flies perform normally in our food-seeking assay.

[Editors' note: the author responses to the re-review follow.]

The manuscript has been improved but there are some remaining changes to the text that need to be addressed before acceptance, as outlined below:1) Based on RNAi experiments, the authors concluded that "DANs receive surprisingly rich inputs of hunger and satiety signals". To support the conclusion, authors would need to show a) which specific serotonergic or peptidergic neurons mediate hunger/satiety signals, b) they are anatomically and functionally connected to DANs, c) localize receptors in DAN's dendrites, and d) establish physiological effect of serotonin/neuropeptides on DANs. Behavioral phenotype with RNAi may provide good starting point for such endeavor, but I would describe it as a modest evidence for "rich inputs of hunger and satiety signals". Please tone down the conclusions and discuss the caveats of these experiments.

We have toned down our conclusions (Abstract and Introduction, last paragraph) and discussed the caveats (subsection “The MB potentially receives rich inputs of hunger and satiety signals”).

2) Related to the above criticism, GABA receptor knock down in Figure 6B-D is consistent with the model in Figure 6F, but does not necessarily prove it. It is unclear if the RNAi knock down phenotype is due to disruption of GABA signals from MB-MVP2 or other GABAergic neurons including MB-APL. Please note this caveat.

We have noted this caveat in the revised manuscript (subsection “GABAergic inputs in the α1 and β′2 lobe compartments promote yeast food seeking behavior in hungry flies”, last paragraph).

3) The new Syd1-GFP data is possibly interesting, but it is very weak evidence for state dependent modulation of active zones. Syd1-GFP is under the control of GAL4/UAS rather than endogenous promoter as in brp-GFP BAC (Chen et al., 2014). Therefore GFP signals do not necessarily correlate with amount of endogenous Syd1. Also Syd1-GFP fluorescence would not change depending on whether Syd1-GFP molecules are in the functional active zone molecular complex or outside of active zone as a non-functional form. Authors would need more direct measurement of active zones quantity and density. Please state this caveat.

We have stated this caveat in the revised manuscript (subsection “Physiological properties of the DANs are modulated by starvation”, last paragraph).

4) In Figure 5 and 9, it would be more intuitive if color for fed and starved are swapped. The color for "starved" should be more salient than basal "fed" state.

We have swapped the colors in Figure 5 and 9. To make the colors consistent throughout the figures, we have also swapped the colors for fed and hungry in other figures.

5) Figure 14 explains how the MBON that showed reduced response at hunger induces food-seeking behavior, but this can't be generalized to all the MBONs. In other words, it looks as if there were no other circuit explanations. This figure should be a figure supplement and the authors can also try to shorten the text explaining this model.

We have moved the figure to Figure 12—figure supplement 1 and shortened the text explaining the model.

6) Table 2 shows extreme efforts that the authors have made, but is very hard to follow. Please make it simpler by showing the results by the cell types but not drivers.

We tried to display the table by the cell types, but it still looks complicated. Consequently, we have completely redrafted the table and think that it is now simpler and easier to follow.

7) The newly added Figure 4 has too much information. Reconsider the essential dataset to show in the main figure and move the others (e.g. the permissive control to a figure supplement).

We have moved the permissive controls to Figure 4—figure supplement 1. Following the reviewers’ suggestion of being more selective in the results that we show in the main figures, we have moved original Figure 10 to Figure 9—figure supplement 1.